

# The systematic relationships and biogeographic history of ornithischian dinosaurs

Clint A. Boyd

North Dakota Geological Survey, Bismarck, ND, United States

## ABSTRACT

The systematic relationships of taxa traditionally referred to as 'basal ornithopods' or 'hypsilophodontids' remain poorly resolved since it was discovered that these taxa are not a monophyletic group, but rather a paraphyletic set of neornithischian taxa. Thus, even as the known diversity of these taxa has dramatically increased over the past two decades, our knowledge of their placement relative to each other and the major ornithischian subclades remained incomplete. This study employs the largest phylogenetic dataset yet compiled to assess basal ornithischian relationships (255 characters for 65 species level terminal taxa). The resulting strict consensus tree is the most well-resolved, stratigraphically consistent hypothesis of basal ornithischian relationships yet hypothesized. The only non-iguanodontian ornithopod (=basal ornithopod) recovered in this analysis is *Hypsilophodon foxii*. The majority of former 'hypsilophodontid' taxa are recovered within a single clade (Parksosauridae) that is situated as the sister-taxon to Cerapoda. The Parksosauridae is divided between two subclades, the Orodrominae and the Thescelosaurinae. This study does not recover a clade consisting of the Asian taxa *Changchunsaurus*, *Haya*, and *Jeholosaurus* (=Jeholosauridae). Rather, the former two taxa are recovered as basal members of Thescelosaurinae, while the latter taxon is recovered in a clade with *Yueosaurus* near the base of Neornithischia.The endemic South American clade Elasmaria is recovered within the Thescelosaurinae as the sister taxon to *Thescelosaurus*. This study supports the origination of Dinosauria and the early diversification of Ornithischia within Gondwana. Neornithischia first arose in Africa by the Early Jurassic before dispersing to Asia before the late Middle Jurassic, where much of the diversification among non-cerapodan neornithischians occurred. Under the simplest scenario the Parksosauridae originated in North America, with at least two later dispersals to Asia and one to South America. However, when ghost lineages are considered, an alternate dispersal hypothesis has thescelosaurines dispersing from Asia into South America (via North America) during the Early Cretaceous, then back into North America in the latest Cretaceous. The latter hypothesis may explain the dominance of orodromine taxa prior to the Maastrichtian in North America and the sudden appearance and wide distribution of thescelosaurines in North America beginning in the early Maastrichtian. While the diversity of parksosaurids has greatly increased over the last fifteen years, a ghost lineage of over 40 myr is present between the base of Parksosauridae and Cerapoda, indicating that much of the early history and diversity of this clade is yet to be discovered. This new phylogenetic hypothesis provides a comprehensive framework for testing further hypotheses regarding evolutionary patterns and processes within Ornithischia.

Corresponding author
Clint A. Boyd, clintboyd@stratfit.org

## INTRODUCTION

In recent years, considerable controversy arose surrounding the systematic relationships of taxa traditionally considered to be basal members of Ornithopoda (i.e., heterodontosaurids and hypsilophodontids). Once considered to be a stable, well supported portion of the ornithischian evolutionary tree (e.g., *Sereno, 1999*), Ornithopoda (sensu *Sereno, 1998*) is now rarely recovered as a monophyletic group in phylogenetic analyses of ornithischian relationships owing to the recovery of heterodontosaurids near the base of Ornithischia (e.g., *Spencer, 2007*; *Butler, Upchurch & Norman, 2008*). That situation prompted *Butler, Upchurch & Norman (2008)* to redefine Ornithopoda (see Table 1), removing heterodontosaurids as internal specifiers, and restricting the contents of Ornithopoda to only those taxa more closely related to Iguanodontia than to Marginocephalia. Despite this attempt to provide stability to use of the taxon name Ornithopoda, the exact contents of the clade remain poorly understood (*Liu, 2004*; *Butler, 2005*; *Butler, Upchurch & Norman, 2008*; *Boyd et al., 2009*).

The recognition of Hypsilophodontidae as a paraphyletic set of taxa (*Scheetz, 1999*; *Butler, Upchurch & Norman, 2008*; *Boyd et al., 2009*; *Brown, Boyd & Russell, 2011*) raised the question of whether all of these taxa belong within Ornithopoda (sensu *Butler, Upchurch & Norman, 2008*), or if some represent non-cerapodan, basal neornithischian taxa. Efforts to address this question via phylogenetic analyses have proven extremely difficult, with the position of former 'hypsilophodontid' taxa remaining fluid between analyses, with little consensus reached (e.g., *Scheetz, 1999*; *Weishampel et al., 2003*; *Butler, 2005*). Given the high level of confusion regarding their systematic position, some authors (i.e., *Boyd et al., 2009*) chose conservatively to refer to all non-marginocephalian, non-iguanodontian neornithischian taxa as 'basal neornithischians' until this question is adequately addressed. However, the majority of researchers continue to refer to these taxa as basal ornithopods or basal cerapodans, despite the fact that the use of those names implies resolved placement of taxa relative to Marginocephalia that is lacking in most recent phylogenetic analyses of ornithischian relationships (e.g., *Butler, Upchurch & Norman, 2008*; *Butler et al., 2011*; *Makovicky et al., 2011*).

Given the difficulties outlined above, phylogenetic analyses of basal ornithischian and/or neornithischian relationships tended to include few basal neornithischian taxa (e.g., *Spencer, 2007*), focusing on the most complete, well known taxa (e.g., *Hypsilophodon*) and ignoring less complete, but morphologically informative taxa (e.g., *Zephyrosaurus*). Moreover, smaller scale analyses of basal neornithischian relationships tended to include a noticeable level of geographic bias among the included taxa. For example, the dataset published by *Scheetz (1999)* and its subsequent modifications (e.g., *Varricchio, Martin & Katsura, 2007*; *Boyd et al., 2009*; *Brown, Boyd & Russell, 2011*) largely sample North American neornithischian taxa, with a few Asian taxa included. Along those same

Boyd (2015), *PeerJ*, DOI 10.7717/peerj.1523

**Table 1** Phylogenetic definitions used in this study.

| Clade name | Phylogenetic definition | Diagnosis type | Original author | Definition used |
|---|---|---|---|---|
| Ankylopollexia | *Camptosaurus dispar* (*Marsh, 1879*), *Parasaurolophus walkeri Parks, 1922*, their most recent common ancestor and all descendants. | Node | *Sereno, 1986* | *Sereno, 2005* |
| Cerapoda | *Parasaurolophus walkeri Parks, 1922*, *Triceratops horridus Marsh, 1889*, their most recent common ancestor and all descendants. | Node | *Sereno, 1986* | *Butler, Upchurch & Norman, 2008* |
| Clypeodonta | *Hypsilophodon foxii Huxley, 1869*, *Edmontosaurus regalis Lambe, 1917*, their most recent common ancestor, and all of its descendants. | Node | *Norman, 2015* | *Norman, 2015* |
| Dinosauria | *Triceratops horrid-us Marsh, 1889*, *Passer domestics* (*Linnaeus, 1758*), their most recent common ancestor and all descendants. | Node | *Owen, 1842* | *Butler, Upchurch & Norman, 2008* |
| Dryomorpha | *Dryosaurus altus* (*Marsh, 1878*), *Parasaurolophus walkeri Parks, 1922*, their most recent common ancestor and all descendants. | Node | *Sereno, 1986* | This study |
| Dryosauridae | All iguanodontians more closely related to *Dryosaurus altus* (*Marsh, 1878*) than to *Parasaurolophus walkeri Parks, 1922*. | Stem | *Milner & Norman, 1984* | *Sereno, 2005* |
| Elasmaria | *Talenkauen santacrucensis* Novas, Cambiaso, and Ambrosia 2004 and *Macrogryphosaurus gondwanicus Calvo, Porfiri & Novas, 2007*, their most recent common ancestor and all descendants. | Node | *Calvo, Porfiri & Novas, 2007* | *Calvo, Porfiri & Novas, 2007* |
| Euiguanodontia | *Gasparinisaura cincosaltensis* (*Coria & Salgado, 1996*), Dryosauridae (*Milner & Norman, 1984*), Ankylopollexia (*Sereno, 1986*), their most recent common ancestor and all descendants. | Node | *Coria & Salgado, 1996* | *Coria & Salgado, 1996* |
| Euornithopoda | All ornithischians more closely related to *Parasaurolophus walkeri* (*Parks, 1922*) than to *Heterodontosaurus tucki Crompton & Charig, 1962*, *Pachycephalosaurus wyomingensis* (*Gilmore, 1931*), *Triceratops horridus Marsh, 1889*, or *Ankylosaurus marginventris Brown, 1908*. | Stem | *Sereno, 1986* | *Sereno, 2005* |

**Table 1** (*Continued*)

| Clade name | Phylogenetic definition | Diagnosis type | Original author | Definition used |
|---|---|---|---|---|
| Genasauria | *Ankylosaurus magniventris Brown, 1908*, *Stegosaurus stenops Marsh, 1877a*, *Parasaurolophus walkeri Parks, 1922*, *Triceratops horridus* (*Marsh, 1889*), *Pachycephalosaurus wyomingensis* (*Gilmore, 1931*), their most recent common ancestor and all descendants. | Node | *Sereno, 1986* | *Butler, Upchurch & Norman, 2008* |
| Heterodontosauridae | All ornithischians more closely related to *Heterodontosaurus tucki Crompton & Charig, 1962* than to *Parasaurolophus walkeri Parks, 1922*, *Pachycephalosaurus wyomingensis* (*Gilmore, 1931*), *Triceratops horridus Marsh, 1889*, or *Ankylosaurus marginventris Brown, 1908*. | Stem | *Romer, 1966* | *Sereno, 2005* |
| Hypsilophodontidae | All neornithischians more closely related to *Hypsilophodon foxii Huxley, 1869* than to *Parasaurolophus walkeri Parks, 1922*. | Stem | *Dollo, 1882* | *Sereno, 2005* |
| Iguanodontia | All ornithopods more closely related to *Parasaurolophus walkeri Parks, 1922* than to *Hypsilophodon foxii Huxley, 1869* or *Thescelosaurus neglectus Gilmore, 1913*. | Stem | *Dollo, 1888* | *Sereno, 2005* |
| Marginocephalia | *Triceratops horridus Marsh, 1889*, *Pachycephalosaurus wyomingensis* (*Gilmore, 1931*), their most recent common ancestor and all descendants. | Node | *Sereno, 1986* | *Butler, Upchurch & Norman, 2008* |
| Neornithischia | All genasaurians more closely related to *Parasaurolophus walkeri Parks, 1922* than to *Ankylosaurus magniventris Brown, 1908* or *Stegosaurus stenops Marsh, 1877a*. | Stem | *Cooper, 1985* | *Butler, Upchurch & Norman, 2008* |
| Ornithischia | All dinosaurs more closely related to *Triceratops horridus Marsh, 1889* than to either *Passer domesticus* (*Linnaeus, 1758*), or *Saltasaurus loricatus Bonaparte & Powell, 1980*. | Stem | *Seeley, 1887* | *Butler, Upchurch & Norman, 2008* |
| Ornithopoda | All genasaurians more closely related to *Parasaurolophus walkeri Parks, 1922*, than to *Triceratops horridus Marsh, 1889*. | Stem | *Marsh, 1881* | *Butler, Upchurch & Norman, 2008* |
| Orodrominae | All neornithischians more closely related to *Orodromeus makelai Horner & Weishampel, 1988* than to *Thescelosaurus neglectus Gilmore, 1913* or *Parasaurolophus walkeri Parks, 1922*. | Stem | *Brown et al., 2013* | This study |
| Parksosauridae | All neornithischians more closely related to *Parksosaurus warreni Parks, 1926* than to *Hypsilophodon foxii Huxley, 1869*, *Dryosaurus altus* (*Marsh, 1878*), or *Parasaurolophus walkeri Parks, 1922*. | Stem | *Buchholz, 2002* | This study |
**Table 1** (*Continued*)

| Clade name | Phylogenetic definition | Diagnosis type | Original author | Definition used |
|---|---|---|---|---|
| Rhabdodontidae | All iguanodontians more closely related to *Rhabdodon priscus Matheron, 1869* than to *Parasaurolophus walkeri Parks, 1922*. | Stem | *Weishampel et al., 2003* | *Sereno, 2005* |
| Saurischia | All dinosaurs more closely related to *Passer domesticus* (*Linnaeus, 1758*) than to *Triceratops horridus Marsh, 1889*. | Stem | *Seeley, 1887* | *Butler, Upchurch & Norman, 2008* |
| Thescelosauridae | *Thescelosaurus neglectus Gilmore, 1913*, *Orodromeus makelai Horner & Weishampel, 1988*, their most recent common ancestor, and all of its descendants. | Node | *Brown et al., 2013* | *Brown et al., 2013* |
| Thescelosaurinae | All neornithischians more closely related to *Thescelosaurus neglectus Gilmore, 1913* than to *Orodromeus makelai Horner & Weishampel, 1988* or *Parasaurolophus walkeri Parks, 1922*. | Stem | *Sternberg, 1937* | This study |
| Thyreophora | All genasaurians more closely related to *Ankylosaurus magniventris Brown, 1908* than to *Parasaurolophus walkeri Parks, 1922*, *Triceratops horridus Marsh, 1889*, or *Pachycephalosaurus wyomingensis* (*Gilmore, 1931*). | Stem | *Nopcsa, 1915* | *Butler, Upchurch & Norman, 2008* |
lines, analyses of South American taxa tend to heavily sample endemic taxa, while largely ignoring taxa from outside the continent (e.g., *Coria, 1999*; *Novas, Cambiaso & Ambrosio, 2004*; *Calvo, Porfiri & Novas, 2007*). Although these analyses may individually give the impression that the relationships of basal neornithischian taxa are well resolved, in truth the broader interrelationships of these taxa relative to each other and to the major ornithischian subclades (e.g., Marginocephalia) remain ambiguous.

The most extensive analysis of ornithischian relationships yet conducted sought to address, among other issues, the interrelationships of fifteen basal neornithischian taxa (*Butler, Upchurch & Norman, 2008*). That analysis met with limited success, ultimately requiring the incorporation of a combination of less-than-strict consensus methods and the removal of six basal neornithischian taxa to resolve the relationships of the remaining taxa. *Butler, Upchurch & Norman (2008)* conclude their discussion of these 'hypsilophodontid' taxa (their usage) by commenting on the need for further work on the relationships of these important but enigmatic taxa.

During the past decade there was a sharp increase in the number of new basal neornithischian taxa described from across the globe, including new taxa from Asia (e.g., *Zan et al., 2005*; *Huh et al., 2011*; *Makovicky et al., 2011*; *Zheng et al., 2012*), North America (e.g., *Varricchio, Martin & Katsura, 2007*; *Brown, Boyd & Russell, 2011*; *Brown et al., 2013*), South America (*Novas, Cambiaso & Ambrosio, 2004*; *Calvo, Porfiri & Novas, 2007*), and Africa (*Butler, 2005*). Those new taxa provide a wealth of information regarding basal ornithischian evolutionary trends and patterns, though most have yet to be included in a large-scale analysis of basal ornithischian relationships. The aim of this study is to robustly assess basal neornithischian dinosaur relationships using a newly constructed species-level dataset that is the largest yet assembled for this purpose both in the number of terminal taxa and characters. The goals of this study include assessment of the systematic relationships of Australian basal neornithischians, which were never before included in a broad analysis of basal ornithischian relationships, determination of the position of Marginocephalia within Neornithischia to clarify the contents of the clade Ornithopoda, clarification of the interrelationships of those taxa generally referred to as 'hypsilophodontids' and their placement relative to the major ornithischian subclades, and comparison of the results of this analysis to those of other recent phylogenetic analyses of basal ornithischian relationships (i.e., *Buchholz, 2002*; *Spencer, 2007*; *Butler, Upchurch & Norman, 2008*: see Fig. 1). The results of this phylogenetic analysis provide new insight into the evolutionary and biogeographic history of basal ornithischian dinosaurs and broader relationships within the clade. See Table 1 for a list of phylogenetic definitions used in this study.

## MATERIALS AND METHODS

### Dataset construction

The core of this dataset is composed of characters compiled from four prior analyses of neornithischian relationships (*Weishampel & Heinrich, 1992*; *Scheetz, 1999*; *Weishampel et al., 2003*; *Butler, 2005*). The characters from those analyses were first combined into a single dataset totaling 309 characters. Those characters were then analyzed and congruent

characters were combined, character states were assessed and modified when required, and three characters were excluded (characters 53, 58, and 112 of *Scheetz, 1999*), reducing the dataset to 232 characters. Eleven additional characters were added from other published analyses (*Xu et al., 2002*; *Varricchio, Martin & Katsura, 2007*; *Butler, Upchurch & Norman, 2008*; *McDonald et al., 2010*; *Nesbitt et al., 2010*) largely to address relationships amongst outgroup taxa (e.g., Silesauridae) and within ornithischian subclades (e.g., Dryosauridae). Finally, twelve new characters were added based on personal observations. The final dataset consists of 255 characters. Table S1 provides the character descriptions, Table S2 provides the reference sources for each character, Table S3 contains the final data matrix, and Table 2 contains the list of specimens examined and references consulted for each taxon.

## Taxon selection

The purpose of this analysis is to assess the relationships of non-iguanodontian, non-marginocephalian neornithischian taxa (i.e., basal neornithischians). Specifically, all taxa previously included as members of the paraphyletic clades 'Fabrosauridae' and 'Hypsilophodontidae' were sampled, as were basal members of Iguanodontia whose relationships with 'hypsilophodontid' taxa remains ambiguous (e.g., *Gasparinisaura cincosaltensis*). To determine the position of these taxa relative to major ornithischian subclades, basal members of five additional ornithischian clades (see below) were included in the analysis. This approach was chosen rather than coding each clade as a supraspecific terminal taxon because use of species-level exemplars has been shown to increase the accuracy of phylogenetic analyses (*Wiens, 1998*; *Prendini, 2001*), ensuring that the results of this analysis are as accurate as possible. As a result, this study represents the first analysis of basal ornithischian relationships conducted entirely at the species level and analyzed using a single dataset. In addition to the ornithischian taxa included in this analysis, six non-ornithischian outgroup taxa, including three non-dinosaurian taxa, were included to root the tree. In total, sixty-five species level terminal taxa were included in this analysis. Each of these taxa is briefly discussed below.

### Taxa of interest

The twenty-seven taxa discussed below constitute the focus of this investigation.

*Agilisaurus louderbacki* *Peng, 1990.*  The holotype of *Agilisaurus louderbacki* consists of a nearly complete skeleton from the Middle Jurassic lower Shaximiao Formation in Sichuan Province, China. This taxon was originally referred to the Fabrosauridae by *Peng (1990)* and *Peng (1992)*, a clade now recognized as a paraphyletic assemblage of basal ornithischian taxa (*Butler, Upchurch & Norman, 2008*). Phylogenetic analyses have recovered *Agilisaurus* as either a basal euornithopod (*Buchholz, 2002*; *Weishampel et al., 2003*), or as a basal neornithischian (e.g., *Scheetz, 1999*; *Butler, 2005*; *Varricchio, Martin & Katsura, 2007*; *Butler, Upchurch & Norman, 2008*; *Boyd et al., 2009*; *Brown, Boyd & Russell 2011*). Two other basal neornithischian taxa are also known from that portion of the formation: *Hexinlusaurus multidens*, and *Xiaosaurus dashanpensis*. *Hexinlusaurus multidens* was previously considered to represent a second species of *Agilisaurus* by *Peng*

**Table 2  List of references consulted and specimens examined for this analysis.** Specimen numbers in bold indicate casts of that specimen were examined. Specimen numbers in italics indicate specimens for which additional photographs of those specimens were examined, but the specimen was not examined first hand.

| Taxon | Age | References | Specimens examined |
|---|---|---|---|
| *Abrictosaurus* | Hettangian-Sinemurian | *Thulborn, 1974*; *Hopson, 1975* | |
| *Agilisaurus* | Bathonian-Callovian | *Peng, 1992*; *Barrett, Butler & Knoll, 2005* | |
| *Anabisetia* | Cenomanian-Turonian | *Coria & Calvo, 2002*; *Ibiricu et al., 2010* | |
| *Archaeoceratops* | Aptian-Albian | *You & Dodson, 2003* | IVPP V11114, V11115 |
| *Asilisaurus* | Anisian | *Nesbitt et al., 2010* | |
| *Atlascopcosaurus* | Albian | *Rich & Rich, 1989*; *Agnolin et al., 2010* | **NMV P186153** |
| *Callovosaurus* | Callovian | *Galton, 1980*; *Ruiz-Omenaca, Suberbiola & Galton, 2006* | |
| *Camptosaurus* | Kimmeridgian-Tithonian | *Norman, 2004* | |
| *Changchunsaurus* | Aptian-Cenomanian | *Zan et al., 2005*; *Jin et al., 2010*; *Butler et al., 2011* | |
| *Dryosaurus* | Kimmeridgian-Tithonian | *Galton, 1977*; *Galton, 1981*; *Galton, 1983* | |
| *Dysalotosaurus* | Kimmeridgian | *Galton, 1977*; *Galton, 1981*; *Galton, 1983* | |
| *Echinodon* | Berriasian | *Galton, 1978*; *Galton, 2007* | |
| *Elrhazosaurus* | Aptian | *Galton & Taquet, 1982*; *Galton, 2009* | |
| *Emausaurus* | Toarcian | *Norman, Witmer & Weishampel, 2004b* | |
| *Eocursor* | Norian-Rhaetian | *Butler, Smith & Norman, 2007*; *Butler, 2010* | *SAM-PK-8025* |
| *Fruitadens* | Tithonian | *Butler et al., 2010* | |
| *Gasparinisaura* | Santonian-Campanian | *Coria & Salgado, 1996*; *Salgado, Coria & Heredia, 1997*; *Coria, 1999*; *Coria & Calvo, 2002*; *Ibiricu et al., 2010* | |
| *Haya* | Santonian | *Makovicky et al., 2011* | IGM 100/2017, 100/2014, 100/2016 |
| *Herrerasaurus* | Carnian | *Novas, 1993*; *Sereno, 1993*; *Sereno & Novas, 1993* | |
| *Heterodontosaurus* | Hettangian-Sinemurian | *Crompton & Charig, 1962*; *Santa Luca, Crompton & Charig, 1976*; *Santa Luca, 1980*; *Butler, Porro & Norman, 2008* | *SAM-PK-K337, 1332* |
| *Hexinlusaurus* | Bathonian-Callovian | *He & Cai, 1984*; *Barrett, Butler & Knoll, 2005* | |
| *Hypsilophodon* | Barremian-Aptian | *Galton, 1974a* | |
| *Iguanodon* | Valanginian-Albian | *Norman, 2004* | |
| *Jeholosaurus* | Barremian | *Xu, Wang & You, 2000*; *Barrett & Han, 2009* | IVVP V 12529, IVPP V 15718; PKUP V 1061, 1062, 1063, 1064 |
| **Kaiparowits Orodromine** | Campanian | *Gates et al., 2013* | UMNH VP 12665, 12677, 16281, 16772, 16773, 19470, 21091-21099, 21101-21107 |
| *Koreanosaurus* | Santonian-Campanian | *Huh et al., 2011* | |

**Table 2** (*Continued*)

| Taxon | Age | References | Specimens examined |
|---|---|---|---|
| *Leaellynasaura* | Albian | *Rich & Rich, 1989*; *Agnolin et al., 2010*; *Rich, Galton & Vickers-Rich, 2010* | **NVM P186047** |
| *Lesothosaurus* | Hettangian-Sinemurian | *Galton, 1978*; *Sereno, 1991*; *Knoll, 2002a*; *Knoll, 2002b*; *Butler, 2005* | *SAM-PK-401, 1106* |
| *Liaoceratops* | Barremian | *Xu et al., 2002*; *Xu et al., 2006* | IVPP V12738; V12633 |
| *Lycorhinus* | Hettangian-Sinemurian | *Haughton, 1924*; *Gow, 1975*; *Hopson, 1975*; *Gow, 1990* | |
| *Macrogryphosaurus* | Coniacian | *Calvo, Porfiri & Novas, 2007*; *Ibiricu et al., 2010* | |
| *Marasuchus* | Ladinian | *Sereno & Arcucci, 1994* | |
| *Micropachycephalosaurus* | Campanian | *Dong, 1978*; *Butler & Zhao, 2009* | |
| *Muttaburrasaurus* | Albian | *Bartholomai & Molnar, 1981*; *Molnar, 1996* | |
| *Notohypsilophodon* | Cenomanian-Coniacian | *Martinez, 1998*; *Ibiricu et al., 2010* | |
| *Orodromeus* | Campanian | *Scheetz, 1999* | MOR 294, 403, 473, 1136, 1141; PU 23246, 23442 |
| *Oryctodromeus* | Cenomanian | *Varricchio, Martin & Katsura, 2007*; *Krumenacker, 2010* | BYU 19342, 19347; MOR 1636a, 1636b |
| *Othnielosaurus* | Kimmeridgian-Tithonian | *Galton & Jensen, 1973*; *Galton, 1977*; *Galton, 1978*; *Galton, 1983*; *Galton, 2007* | BYU ESM-163R; UW 24823 |
| *Ouranosaurus* | Aptian | *Norman, Witmer & Weishampel, 2004b* | |
| *Parksosaurus* | Maastrichtian | *Parks, 1926*; *Galton, 1973* | ROM 804 |
| *Pisanosaurus* | Carnian | *Casamiquela, 1967*; *Bonaparte, 1976*; *Gow, 1981*; *Irmis, Parker & Nesbitt, 2007* | |
| *Qantassaurus* | Albian | *Rich & Vickers-Rich, 1999*; *Agnolin et al., 2010* | **NMV P198962, P199075** |
| *Rhabdodon* | Santonian-Maastrichtian | *Garcia et al., 1999*; *Pincemaille-Quillevere, Buffetaut & Quillevere, 2006* | |
| *Sanjuansaurus* | Carnian | *Alcober & Martinez, 2010* | |
| *Scelidosaurus* | Sinemurian | *Norman, Witmer & Weishampel, 2004b* | |
| *Scutellosaurus* | Hettangian-Sinemurian | *Colbert, 1981*; *Rosenbaum & Padian, 2000* | TMM 43647.7, 43663.1, 43664.1, 43687.16 |
| *Silesaurus* | Carnian | *Dzik, 2003*; *Piechowski & Dzik, 2010* | |
| *Stenopelix* | Berriasian | *Butler & Sullivan, 2009* | |
| *Stormbergia* | Hettangian-Sinemurian | *Butler, 2005* | *SAM-PK-1105* |
| *Talenkauen* | Maastrichtian | *Novas, Cambiaso & Ambrosio, 2004*; *Ibiricu et al., 2010* | |
| *Tawa* | 213–215 myr | *Nesbitt et al., 2009* | |
| *Tenontosaurus dossi* | Aptian | *Winkler, Murry & Jacobs, 1997* | |
| *Tenontosaurus tilletti* | Aptian-Albian | *Forster, 1990* | |
| *Thescelosaurus assiniboiensis* | Maastrichtian | *Galton, 1989*; *Galton, 1997*; *Brown, Boyd & Russell, 2011* | RSM P 1225.1 |

**Table 2** (*Continued*)

| Taxon | Age | References | Specimens examined |
|---|---|---|---|
| *Thescelosaurus garbanii* | Maastrichtian | *Morris, 1976* | |
| *Thescelosaurus neglectus* | Maastrichtian | *Gilmore, 1913*; *Gilmore, 1915*; *Sternberg, 1940*; *Galton, 1974b*; *Galton, 1995*; *Galton, 1997*; *Galton, 1999*; *Morris, 1976*; *Boyd et al., 2009* | NCSM 15728; USNM 7757, 7758 |
| *Tianyulong* | Oxfordian-Kimmeridgian | *Zheng et al.,2009*; *Gao & Shubin, 2012* | |
| *Valdosaurus* | Barriasian-Barremian | *Barrett et al., 2011* | |
| *Wannanosaurus* | Campanian-Maastrichtian | *Hou, 1977*; *Butler & Zhao, 2009* | |
| *Yandusaurus* | Oxfordian | *He, 1979*; *He & Cai, 1984*; *Barrett, Butler & Knoll, 2005* | |
| *Yinlong* | Oxfordian | *Xu et al., 2006* | IVPP V14530 |
| *Yueosaurus* | Albian-Cenomanian | *Zheng et al., 2012* | |
| *Zalmoxes robustus* | Maastrichtian | *Weishampel et al., 2003* | |
| *Zalmoxes shqiperorum* | Maastrichtian | *Weishampel et al., 2003*; *Godefroit, Codrea & Weishampel, 2009* | |
| *Zephyrosaurus* | Aptian | *Sues, 1980*; *Kutter, 2004* | YPM 56695 |

*(1990)* and *Peng (1992)* based on several shared characters, but a reassessment of this referral by *Barrett, Butler & Knoll (2005)* found this referral to be unwarranted.

*Anabisetia saldiviai Coria & Calvo, 2002.* The South American taxon *Anabisetia saldiviai* is known from the Late Cretaceous Lisandro Formation of Argentina and is based upon a partial skull and postcranial skeleton. It was first reported by *Coria (1999)*, but was not formally named and described until 2002 by *Coria & Calvo (2002)*. This taxon has been recovered as either an euiguanodontian (e.g., *Coria, 1999*; *Coria & Calvo, 2002*) or as a basal iguanodontian (e.g., *Butler, Upchurch & Norman, 2008*).

*Atlascopcosaurus loadsi Rich & Rich, 1989.* The Australian taxon *Atlascopcosaurus loadsi*, from the Early Cretaceous Eumeralla Formation (Otway Group: *Agnolin et al., 2010*), is based upon the holotype maxilla and a few referred specimens including isolated teeth, a maxilla, and dentaries (*Rich & Rich, 1989*). This taxon was originally referred to the Hypsilophodontidae, and most subsequent treatments accepted that referral (e.g., *Rich & Vickers-Rich, 1999*). *Norman et al. (2004)* suggested that *Atlascopcosaurus* may be closely related to the South American taxon *Gasparinisaura. Agnolin et al. (2010)* referred to *Atlascopcosaurus* as a non-dryomorph ornithopod that shares many features in common with both *Gasparinisaura* and another South American taxon, *Anabisetia*, perhaps indicating that these three taxa share a close phylogenetic relationship. However, the latter author also considered *Atlascopcosaurus* to be a *nomen dubium*, though the exact reasons for that referral are not discussed. This study considered *Atlascopcosaurus* to be a diagnosably distinct taxon, and it was retained in the phylogenetic analysis.

*Changchunsaurus parvus Zan et al., 2005.* *Changchunsaurus parvus* is based on a single specimen consisting of a complete skull with partial postcranial skeleton recovered from the 'middle' Cretaceous Quantou Formation of Jilin Province, China. The anatomy of the holotype was recently redescribed and its systematic relationships were analyzed for the first time (*Jin et al., 2010*; *Butler et al., 2011*). Additionally, this taxon was included in an analysis of the systematic relationships of the Asian taxon *Haya griva* (*Makovicky et al., 2011*). *Changchunsaurus parvus* was recovered by *Butler, Upchurch & Norman (2008)* and *Makovicky et al. (2011)* near the base of Ornithopoda as the sister taxon to *Jeholosaurus shangyuanensis*, another small-bodied taxon from the Early Cretaceous of China.

*Gasparinisaura cincosaltensis Coria & Salgado, 1996.* The holotype of *Gasparinisaura cincosaltensis* is a nearly complete skull and partial postcranial skeleton from the Late Cretaceous Rio Colorado Formation of Argentina. Additional material that provided more information regarding the postcranial anatomy of this taxon was referred to this taxon by *Salgado, Coria & Heredia (1997)*. Considerable controversy surrounds the phylogenetic position of this taxon, with various hypotheses placing it as a hypsilophodontid (e.g., *Butler, 2005*), a basal euornithopod (*Weishampel et al., 2003*), a basal iguanodontian (e.g., *Scheetz, 1999*; *Varricchio, Martin & Katsura, 2007*; *Boyd et al., 2009*), or as an euiguanodontid (e.g., *Coria & Salgado, 1996*; *Salgado, Coria & Heredia, 1997*).

*Haya griva Makovicky et al., 2011.* The holotype and referred specimens of *Haya griva* preserve representative portions of nearly the entire skeleton. This taxon was recovered from the Khugenetslavkant locality within the Late Cretaceous Javkhlant Formation of Mongolia. The phylogenetic analysis conducted by *Makovicky et al. (2011)*, which used the dataset published by *Butler et al. (2011)*, recovered *H. griva* as the sister taxon to a clade consisting of the Asian taxa *Jeholosaurus shangyuanensis* and *Changchunsaurus parvus*. In the strict consensus tree, the clade containing those three taxa was recovered in a polytomy at the base of Neornithischia. *Han et al. (2012)* also recovered those three taxa in a clade, which they named Jeholosauridae, and suggested that the Asian taxa *Koreanosaurus* and *Yueosaurus* may also belong to that clade.

*Hexinlusaurus multidens (He & Cai, 1983).* *Hexinlusaurus multidens* is known from the nearly complete holotype, lacking only the anterior-most portion of the skull, most of the mandibles, and the distal portion of the tail, as well as a second, disarticulated specimen (*He & Cai, 1984*). It was recovered from the Middle Jurassic lower Shaximiao Formation of Sichuan Province, China. The species was originally referred to the taxon *Yandusaurus* (*He & Cai, 1983*; *He & Cai, 1984*), but subsequent authors referred it to either *Othnielosaurus* (e.g., *Paul, 1996*) or the contemporaneous taxon *Agilisaurus* (e.g., *Peng, 1990*; *Peng, 1992*). A recent review of the morphology and taxonomy of the species by *Barrett, Butler & Knoll (2005)* led them to erect a new taxon for this species, *Hexinlusaurus*. *Hexinlusaurus multidens* was included in many prior cladistic analyses of ornithischian relationships, though it was usually labeled as *Yandusaurus* (e.g., *Weishampel & Heinrich, 1992*; *Scheetz, 1999*). Regardless of its designation, it is recovered as a basal member of either Hypsilophodontidae (e.g., *Weishampel & Heinrich, 1992*), Euornithopoda (e.g., *Buchholz, 2002*), or Neornithischia (e.g., *Scheetz, 1999*; *Butler, 2005*; *Varricchio, Martin & Katsura, 2007*; *Butler, Upchurch & Norman, 2008*; *Boyd et al., 2009*).

*Hypsilophodon foxii Huxley, 1869.* *Hypsilophodon foxii* was the first discovered and one of the best known taxa traditionally referred to the Hypsilophodontidae. Multiple specimens preserving representative portions of the entire skeleton are known from the Early Cretaceous Wessex Formation of England. Despite being the internal specifier for the clade Hypsilophodontidae (*Sereno, 2005*), its systematic position with respect to other taxa traditionally referred to Hypsilophodontidae remains ambiguous, with some analyses recovering it as the sole member of the clade (e.g., *Scheetz, 1999*; *Buchholz, 2002*; *Weishampel et al., 2003*; *Varricchio, Martin & Katsura, 2007*; *Boyd et al., 2009*), while others recover at least a reduced version of a monophyletic Hypsilophodontidae (e.g., *Butler, 2005*).

*Jeholosaurus shangyuanensis Xu, Wang & You, 2000.* *Jeholosaurus shangyuanensis* is a small-bodied taxon from the Early Cretaceous Yixian Formation of the Liaoning Province in China. The holotype and paratype specimens largely preserve only cranial material, and the cranial anatomy of this taxon was recently redescribed in detail based on the discovery of additional referred specimens (*Barrett & Han, 2009*). The postcranial anatomy of this taxon remained poorly known until additional material was described by *Han et al. (2012)*. The systematic position of the taxon remains poorly resolved, with the most extensive

analysis of ornithischian relationships placing it in an unresolved position at the base of Ornithopoda (*Butler, Upchurch & Norman, 2008*), though the analysis of *Han et al. (2012)* recovered *Jeholosaurus* in a clade with *Changchunsaurus parvus* and *Haya griva* that they named Jeholosauridae. The analysis presented herein incorporates unpublished data from several undescribed specimens of *J. shangyuanensis* curated at Peking University in Beijing, China that consist of articulated cranial and postcranial skeletons that provide new insights into the phylogenetic position of *J. shangyuanensis*.

*Kaiparowits orodromine (Gates et al., 2013).* The 'Kaiparowits orodromine' is a small-bodied taxon from the Late Cretaceous Kaiparowits Formation of Utah (*Gates et al., 2013*). The best specimen preserves fragmentary cranial and postcranial elements from an immature individual that preserves autapomorphic traits that make it diagnosably distinct from all other known ornithischian taxa (*Boyd, 2012*). Several other presumably juvenile specimens are also referable to this taxon, including an articulated manus from one individual and a set of left and right pedes from another individual, providing insight into the morphology of much of the postcranial skeleton.

*Koreanosaurus boseongensis Huh et al., 2011.* *Koreanosaurus boseongensis* is based upon two partially articulated postcranial skeletons, designated as the holotype and paratype, and a third specimen consisting of a fragmentary hind limb. All of these specimens were recovered from the Late Cretaceous Seonso Conglomerate of South Korea. *Koreanosaurus boseongensis* was tentatively referred to the Ornithopoda by *Huh et al. (2011)* as the sister taxon to *Orodromeus*; however, no phylogenetic analysis was conducted in the original publication (*Huh et al., 2011*).

*Leaellynasaura amicagraphica Rich & Rich, 1989.* *Leaellynasaura amicagraphica*, from the Early Cretaceous Eumeralla Formation (Otway Group: *Agnolin et al., 2010*) of Australia, is known from a holotype specimen (partial left portion of a skull) and several referred specimens. Three of the referred specimens were found at the same locality as the holotype specimen, and it was argued repeatedly that all of these specimens belong to the holotype individual (*Rich & Rich, 1989*; *Rich, Galton & Vickers-Rich, 2010*) based upon analysis of the original site map and the fact that several of the blocks containing these fossils interlock with each other. *Leaellynasaura amicagraphica* was originally assigned to the Hypsilophodontidae (*Rich & Rich, 1989*), and others have referred to it as a non-iguanodontian ornithopod positioned more basally than *Gasparinisaura* (*Agnolin et al., 2010*), which is consistent with the original referral. However, others argued that it is a non-dryomorph iguanodontian (*Herne & Salisbury, 2009*). The most recent assessment of the anatomy of *Leaellynasaura* outlined character evidence supporting the latter taxonomic placement, though only referred the taxon to Ornithopoda (*Rich, Galton & Vickers-Rich, 2010*).

*Lesothosaurus diagnosticus Galton, 1978.* *Lesothosaurus diagnosticus* is a small-bodied taxon from the ''Red Beds'' of the Early Jurassic Upper Elliot Formation, southern Africa. Numerous specimens are referred to this taxon (see *Butler (2005)* for a review) and together

they preserve much of the cranial and postcranial skeleton. The systematic position of *Lesothosaurus* remains contentious; it is hypothesized as either a basal ornithischian (e.g., *Norman, Witmer & Weishampel, 2004a*), a basal neornithischian (e.g., *Scheetz, 1999*; *Varricchio, Martin & Katsura, 2007*; *Butler, 2005*; *Boyd et al., 2009*) or a basal thyreophoran (*Spencer, 2007*; *Butler, Upchurch & Norman, 2008*). Clarifying the relationships of this taxon is a key step to understanding the evolutionary history of Ornithischia. Controversy also surrounds the taxonomic diversity of non-heterodontosaurid ornithischian taxa from the Upper Elliot Formation. *Butler (2005)* recognized the presence of two taxa, *L. diagnosticus* and *Stormbergia dangershoeki*. Other authors (e.g., *Knoll, 2002a*; *Knoll, 2002b*) argued that the material referred to *S. dangershoeki* actually represents the adult form of *L. diagnosticus*, and histological evidence consistent with that interpretation was presented by *Knoll, Padian & de Ricqles (2010)*. Neither of these taxa preserve distinct autapomorphies; rather, they are differentiated based upon unique combinations of character states, many of which are plesiomorphic for Ornithischia (*Butler, 2005*). However, because the synonymy of these taxa is not yet formally proposed, *L. diagnosticus* and *S. dangershoeki* are treated as distinct taxa in this study.

*Macrogryphosaurus gondwanicus Calvo, Porfiri & Novas, 2007*. *Macrogryphosaurus gondwanicus* is a large-bodied taxon from the Late Cretaceous Portezuelo Formation of Argentina. The holotype and only known specimen consists of an incomplete postcranial skeleton preserving almost the entire vertebral column with associated cervical and dorsal ribs, both pelvic girdles, a sternal plate, and four intercostal plates (*Calvo, Porfiri & Novas, 2007*). *Macrogryphosaurus gondwanicus* was previously recovered as a basal euiguanodontian and the sister taxon to *Talenkauen santacrucensis* (*Calvo, Porfiri & Novas, 2007*). A new clade, Elasmaria *Calvo, Porfiri & Novas, 2007*, was erected by *Calvo, Porfiri & Novas (2007)* to contain these two taxa. The most recent analysis of this taxon's systematic relationships recovers it in an unresolved position within a more inclusive Elasmaria that contains several other Gondwanan taxa (*Rozadilla et al., 2016*).

*Notohypsilophodon comodorensis Martinez, 1998*. *Notohypsilophodon comodorensis* is based on a partial postcranial skeleton from the Late Cretaceous Bajo Barreal Formation in Argentina. Described as the first hypsilophodontid recognized from South America, it was previously recovered either as an unresolved position at the base of Ornithopoda (*Coria, 1999*) or as an ornithopod within the clade Elasmaria along with several other Gondwanan taxa (*Rozadilla et al., 2016*).

*Orodromeus makelai Horner & Weishampel, 1988* *Orodromeus makelai* was briefly described by *Horner & Weishampel (1988)* based on a nearly complete skull and postcranial skeleton from the Late Cretaceous, upper Two Medicine Formation of Montana. Numerous specimens from that formation are referred to this taxon, and its anatomy is relatively well known. However, the most extensive descriptive work on this taxon completed to date is an unpublished dissertation (*Scheetz, 1999*), though additional accounts of the long bone histology of this taxon were published (*Horner et al., 2009*). In phylogenetic analyses, *O. makelai* is consistently recovered as the sister taxon of *Zephyrosaurus schaffi* (e.g., *Scheetz,*

*1999*; *Buchholz, 2002*; *Varricchio, Martin & Katsura, 2007*), though the placement of those two taxa within Neornithischia remains problematic.

*Oryctodromeus cubicularis Varricchio, Martin & Katsura, 2007.* *Oryctodromeus cubicularis* was originally described based on a presumed adult holotype (premaxillae, partial braincase, and postcranial elements) and a paratype consisting of disarticulated cranial and postcranial elements from at least two immature individuals, all recovered from a single locality within the early Late Cretaceous Blackleaf Formation of Montana. Subsequently, additional material referable to this taxon was described from the contemporaneous Wayan Formation of Idaho (*Krumenacker, 2010*), which extends the geographical range of *Oryctodromeus* and adds to our knowledge of its anatomy. This taxon is always recovered within a clade along with *Orodromeus makelai* and *Zephyrosaurus schaffi* (e.g., *Varricchio, Martin & Katsura, 2007*; *Brown et al., 2013*). More recently, a new taxon from the upper Oldman Formation of Alberta, *Albertadromeus syntarsus Brown et al., 2013*, was also recovered within that clade, which is now named Orodrominae (*Brown et al., 2013*).

*Othnielosaurus consors (Marsh, 1894).* The holotype of *Othnielia rex* (*Marsh, 1877b*) is a left femur that preserves no autapomorphies; thus, it was declared a *nomen dubium* by *Galton (2007)*. A partial, articulated skeleton previously referred to this taxon, BYU ESM-163R from the Upper Jurassic Morrison Formation of North America, was erected as the holotype of a new taxon, *Othnielosaurus consors*, and all material previously referred to *Othnielia rex* is now referred to *O. consors*. *Galton (1973)* originally referred BYU ESM-163R to the Hypsilophodontidae. Phylogenetic analyses recovered *O. consors* (or the conspecific *O. rex*) as closely related to the Asian basal neornithischian taxa *Agilisaurus louderbacki*, *Hexinlusaurus multidens*, and *Yandusaurus hongheensis* at the base of either Hypsilophodontidae (e.g., *Weishampel & Heinrich, 1992*), Euornithopoda (e.g., *Buchholz, 2002*), or Neornithischia (e.g., *Scheetz, 1999*; *Varricchio, Martin & Katsura, 2007*; *Butler, Upchurch & Norman, 2008*; *Boyd et al., 2009*).

*Parksosaurus warreni (Parks, 1926).* An articulated specimen preserving a partial skull and relatively complete postcranial skeleton was discovered in the Late Cretaceous Tolman Member of the Horseshoe Canyon Formation (Edmonton Group) of Alberta, Canada (*Eberth & Braman, 2012*) and was recognized as the holotype of a new species of *Thescelosaurus*, *Thescelosaurus warreni* (*Parks, 1926*). *Sternberg (1937)* and *Sternberg (1940)* subsequently removed this species from *Thescelosaurus* and placed it in its own taxon, *Parksosaurus*. Recent analysis of all specimens previously referred to the taxon *Thescelosaurus* upheld the validity of *Parksosaurus*, finding it to be diagnostically distinct from all specimens previously referred to *Thescelosaurus* (*Boyd et al., 2009*). The systematic placement of *P. warreni* remains uncertain, with phylogenetic analyses hypothesizing it as either the sister taxon to *Gasparinisaura* (e.g., *Buchholz, 2002*; *Butler, Upchurch & Norman, 2008*), *Thescelosaurus* (e.g., *Weishampel et al., 2003*; *Boyd et al., 2009*; *Brown, Boyd & Russell, 2011*), or *Hypsilophodon* (e.g., *Weishampel & Heinrich, 1992*). *Boyd (2014)* argued for a close relationship between *Parksosaurus* and *Thescelosaurus* based on examination

of the most complete skull yet referred to *Thescelosaurus neglectus* and of the recently re-prepared holotype skull of *P. warreni*, but that hypothesis was not tested in a phylogenetic analysis. Those new character observations are included in this study.

*Qantassaurus intrepidus Rich & Vickers-Rich, 1999.* *Qantassaurus intrepidus*, from the Early Cretaceous Wonthaggi Formation (Strzelecki Group: *Agnolin et al., 2010*) of Australia, is known from the holotype dentary and two referred dentaries, which are diagnosed by their relatively short anteroposterior length compared to their dorsoventral thickness. *Qantassaurus intrepidus* originally was referred to the Hypsilophodontidae (*Rich & Vickers-Rich, 1999*), and was also considered a non-dryomorph ornithopod (*Agnolin et al., 2010*), though those two statements are not mutually exclusive considering 'hypsilophodontids' were traditionally placed at the base of Ornithopoda, below the clade Dryomorpha (e.g., *Butler, Upchurch & Norman, 2008*). The systematic relationships of *Qantassaurus* were never investigated in a phylogenetic analysis prior to this study.

*Stormbergia dangershoeki Butler, 2005.* All specimens of *Stormbergia dangershoeki* are from the 'Red Beds' of the Lower Jurassic upper Elliot Formation of southern Africa. The holotype and paratype are partial postcranial skeletons. Although some authors considered these specimens to represent a valid taxon (e.g., *Butler, 2005*; *Butler, Upchurch & Norman, 2008*), others argue that the morphological differences noted between *Stormbergia dangershoeki* and the contemporaneous *Lesothosaurus diagnosticus* are a result of ontogenetic variation within a single taxon, with *L. diagnosticus* representing the smaller, presumably juvenile form and *S. dangershoeki* representing the larger, presumably adult form (*Knoll, 2002a*; *Knoll, 2002b*; *Knoll, Padian & de Ricqles, 2010*). There is some support for that hypothesis based on histological evidence (*Knoll, Padian & de Ricqles, 2010*), but further study is needed before the question of the validity of *S. dangershoeki* is satisfactorily answered. Additionally, phylogenetic analyses that include both of these taxa consistently place them in disparate positions within the base of Ornithischia based on the presence of unique combinations of key ornithischian characters in each taxon (e.g., *Butler, Upchurch & Norman, 2008*). Therefore, both *L. diagnosticus* and *S. dangershoeki* are retained as terminal taxa.

*Talenkauen santacrucensis Novas, Cambiaso & Ambrosio, 2004.* The holotype and only known specimen of *Talenkauen santacrucensis* consists of a fragmentary skull and partial postcranial skeleton from the Late Cretaceous Pari Aike Formation in the Santa Cruz Province of Argentina. Prior phylogenetic analyses recovered *Talenkauen* as either a basal euiguanodontian (e.g., *Novas, Cambiaso & Ambrosio, 2004*; *Calvo, Porfiri & Novas, 2007*) or as a basal iguanodontian (e.g., *Butler, Upchurch & Norman, 2008*). In the only phylogenetic analysis that included both *Talenkauen* and the South American taxon *Macrogryphosaurus gondwanicus* (also from the Late Cretaceous of Argentina), these two taxa were recovered as sister taxa and identified as part of a new clade, Elasmaria (*Calvo, Porfiri & Novas, 2007*).

*Thescelosaurus assiniboiensis Brown, Boyd & Russell, 2011.* *Thescelosaurus assiniboiensis* is known from a single specimen consisting of a fragmentary skull and partial postcranial skeleton from the Late Cretaceous Frenchman Formation of Saskatchewan, Canada, though other material from that formation are likely referable to this taxon. Originally referred to the type species of *Thescelosaurus* (*T. neglectus*), the holotype of *T. assiniboiensis* preserves autapomorphic traits that make it diagnosably distinct from all other ornithischian taxa (*Brown, Boyd & Russell, 2011*). Both prior phylogenetic analyses that included this taxon placed it within a *Thescelosaurus* clade as the sister taxon to *Parksosaurus warreni* (*Boyd et al., 2009*; *Brown, Boyd & Russell, 2011*).

*Thescelosaurus garbanii Morris, 1976.* The holotype of *Thescelosaurus garbanii* is a fragmentary postcranial skeleton consisting of a few vertebrae and a partial hind limb from the Hell Creek Formation of Montana. Despite the incomplete nature of this specimen, it preserves an apomorphic structure of the ankle that makes it diagnosably distinct from all other ornithischian taxa. Additionally, *Boyd et al. (2009)* confirmed the referral of this species to the taxon *Thescelosaurus* based upon the preservation of a diagnostic set of character states present in the hind limb, recovering it in a phylogenetic analysis as part of a *Thescelosaurus* clade.

*Thescelosaurus neglectus Gilmore, 1913.* *Thescelosaurus neglectus* is a relatively large-bodied taxon from the Late Cretaceous of North America and is the type species for the taxon *Thescelosaurus*. This taxon is known from numerous specimens, one of which includes a well-preserved, complete skull (*Boyd, 2014*). A recent review of specimens referred to *Thescelosaurus* and other closely related taxa resulted in the synonymization of the contemporaneous taxon *Bugenasaura* with *Thescelosaurus* and confirmed the separation of *Thescelosaurus* and *Parksosaurus* (*Boyd et al., 2009*). The systematic position of *T. neglectus* within Ornithischia remains hotly debated. It was originally thought to be closely related to basal ankylopollexians (i.e., *Camptosaurus dispar*) within Ornithopoda, based on a preliminary examination of the hypodigm material (*Gilmore, 1913*), but was soon after referred to the Hypsilophodontidae (*Gilmore, 1915*). That referral was upheld by most subsequent authors for more than sixty years (e.g., *Parks, 1926*; *Swinton, 1936*; *Janensch, 1955*; *Romer, 1956*; *Romer, 1966*; *Thulborn, 1970*; *Thulborn, 1972*), with a few notable exceptions. *Sternberg (1940)* placed *T. neglectus* in its own clade within Hypsilophodontidae, which he named Thescelosaurinae (=Thescelosauridae of *Sternberg, (1937)*), a referral that was followed by some authors (e.g., *Kuhn, 1966*; *Morris, 1976*). *Galton (1971a*, *1971b*, *1972*, *1973*, *1974b*) argued against the placement of *T. neglectus* within Thescelosaurinae and even Hypsilophodontidae, instead referring the taxon to Iguanodontidae. *Galton (1995)*, *Galton (1997)* and *Galton (1999)* later reassessed that referral and instead assigned *T. neglectus* to the Hypsilophodontidae. Despite these taxonomic disagreements, the placement of *T. neglectus* within Ornithopoda (sensu *Butler, Upchurch & Norman, 2008*) was uncontested by all these authors.

Inclusion of *T. neglectus* in recent phylogenetic analyses of ornithischian relationships brought into question its placement within Ornithopoda (sensu *Butler, Upchurch &*

*Norman, 2008*). Several analyses that included *T. neglectus* do not include marginocephalian taxa, making it impossible to determine if *T. neglectus* is placed within a monophyletic Ornithopoda because they do not offer a strong assessment of ornithopod monophyly (e.g., *Weishampel & Heinrich, 1992*; *Scheetz, 1999*; *Varricchio, Martin & Katsura, 2007*; *Boyd et al., 2009*). Additionally, the strict consensus trees produced by *Butler (2005)*, *Spencer (2007)*, and *Butler, Upchurch & Norman (2008)* placed *T. neglectus* in a large polytomy at the base of Neornithischia, a position that precludes its definitive referral to Ornithopoda. Another published study (*Buchholz, 2002*) did not include the strict consensus tree of the recovered set of ten most parsimonious trees, presenting only one of the recovered most parsimonious trees, making it impossible to determine if *T. neglectus* was recovered within Ornithopoda in all ten of the most parsimonious trees. Finally, *Weishampel et al. (2003)* set their supraspecific terminal taxon Marginocephalia as an outgroup taxon, making the unambiguous recovery of *T. neglectus* within Ornithopoda a certainty. Thus, in no previous phylogenetic analysis of ornithischian relationships was the placement of *T. neglectus* within Ornithopoda unambiguously confirmed (sensu *Butler, Upchurch & Norman, 2008*).

*Yandusaurus hongheensis He, 1979.* *Yandusaurus hongheensis* is based on a fragmentary skull and postcranial skeleton from the Middle Jurassic upper Shaximiao Formation of Sichuan, China. Though this taxon is listed as being included in several prior phylogenetic analyses of ornithischian relationships (e.g., *Weishampel & Heinrich, 1992*; *Scheetz, 1999*), in most of these cases the taxon included was the more complete species 'Yandusaurus' *multidens*, which was subsequently removed from *Yandusaurus* and placed in a new taxon, *Hexinlusaurus* (*Barrett, Butler & Knoll, 2005*). The most recent phylogenetic analysis that included *Y. hongheensis* as a terminal taxon recovered it in an unresolved position within Neornithischia (*Butler, Upchurch & Norman, 2008*).

*Yueosaurus tiantaiensis Zheng et al., 2012.* This taxon is known from a single, fragmentary postcranial skeleton from the Late Cretaceous Liangtoutang Formation of Zhejiang Province, China. Despite the fragmentary nature of the specimen, the presence of three autapomorphies on the scapula confirms the validity of this taxon. The systematic relationships of *Y. tiantaiensis* have never been assessed in a phylogenetic analysis.

*Zephyrosaurus schaffi Sues, 1980.* *Zephyrosaurus schaffi*, a North American taxon from the Early Cretaceous Cloverly Formation, is based upon an incomplete skull and extremely fragmentary postcranial skeleton. Additional material referable to this taxon is known, but remains either undescribed or described only in an unpublished thesis (*Kutter, 2004*), limiting our understanding of the taxon. In phylogenetic analyses, it is frequently recovered as the sister taxon to *Orodromeus makelai* (e.g., *Weishampel & Heinrich, 1992*; *Scheetz, 1999*; *Varricchio, Martin & Katsura, 2007*; *Boyd et al., 2009*) and was recovered within the clade Orodrominae by *Brown et al. (2013)*.

### Basal ornithischian taxa
Two basal ornithischian taxa that are not considered part of the ingroup and do not fall within any of the major ornithischian subclades, but are key for evaluating ornithischian

relationships, were included. These two taxa provide important insights into the early evolution of ornithischian dinosaurs. *Pisanosaurus mertii Casamiquela, 1967* is traditionally considered the most basal ornithischian taxon yet discovered, a hypothesis supported by phylogenetic analyses of ornithischian relationships (e.g., *Butler, 2005*; *Spencer, 2007*; *Butler, Upchurch & Norman, 2008*). *Eocursor parvus Butler, Smith & Norman, 2007* is currently considered a non-genasaurian, ornithischian dinosaur, situated between the clades Heterodontosauridae and Thyreophora (e.g., *Butler, Smith & Norman, 2007*). Alternatively, *Spencer (2007)* recovered *Eocursor* as a basal neornithischian, but still basal to the Heterodontosauridae, which was also placed within Neornithischia.

### Species exemplars of major ornithischian subclades

The following ornithischian taxa were included in this analysis to represent major subclades whose monophyly is supported by prior analyses of ornithischian relationships (e.g., *Butler, Upchurch & Norman, 2008*). Inclusion of species-level exemplars from all four major ornithischian subclades is critical for accurately resolving the relationships of the twenty-seven taxa under study in this analysis and obtaining a clear understanding of character evolution and patterns of biogeographic dispersal within Ornithischia.

*Heterodontosauridae Kuhn, 1966 (Sensu Sereno, 2005).* The phylogenetic position of Heterodontosauridae has been problematic over the past two decades (e.g., *Sereno, 1999*; *Buchholz, 2002*; *Butler, 2005*; *Butler, Upchurch & Norman, 2008*), being hypothesized either within Ornithopoda (e.g., *Sereno, 1999*), as the sister-taxon to Marginocephalia (e.g., *Buchholz, 2002*), near the base of Neornithischia (e.g., *Butler, 2005*), or outside of Genasauria (e.g., *Butler, Upchurch & Norman, 2008*). Regardless of the placement of this clade within Ornithischia, a monophyletic core was consistently recovered. Six taxa were selected to represent this clade: the African taxa *Abrictosaurus consors* (*Thulborn, 1974*), *Heterodontosaurus tucki Crompton & Charig, 1962*, and *Lycorhinus angustidens Haughton, 1924*; the European taxon *Echinodon becklesii Owen, 1861*; the North American taxon *Fruitadens haagarorum Butler et al., 2010*; and, the Asian taxon *Tianyulong confuciusi Zheng et al., 2009*. When combined these taxa represent the full temporal range of this clade.

*Thyreophora Nopsca 1915 (Sensu Butler, Upchurch & Norman, 2008).* The monophyly of Thyreophora is one of the most stable components within Ornithischia (e.g., *Norman, 1984*; *Cooper, 1985*; *Sereno, 1986*; *Sereno, 1999*; *Butler, Upchurch & Norman, 2008*). The European taxa *Emausaurus ernsti Haubold, 1991* and *Scelidosaurus harrisonii Owen, 1861* and the North American taxon *Scutellosaurus lawleri Colbert, 1981* were long recognized as the most basal members of the Thyreophora (e.g., *Sereno, 1999*; *Butler, 2005*; *Butler, Upchurch & Norman, 2008*), and are here included as its representatives. New character data for *S. lawleri* was incorporated from study of additional referred specimens examined by the author and currently under study at The University of Texas at Austin (see Table 2).

*Marginocephalia Sereno, 1986 (Sensu Butler, Upchurch & Norman, 2008).* The monophyly of Marginocephalia was questioned by some researchers (e.g., *Dodson, 1990*; *Sullivan, 2006*), but recent phylogenetic analyses of the Ornithischia all support the monophyly of

this clade (e.g., *Butler, Upchurch & Norman, 2008*). Six marginocephalian taxa whose position within Marginocephalia is confirmed by recent studies (e.g., *Butler, Upchurch & Norman, 2008*) were included in this analysis. These taxa include the ceratopsian dinosaurs *Archaeoceratops oshimai Dong & Azuma, 1997*, *Liaoceratops yanzigouensis Xu et al., 2002*, and *Yinlong downsi Xu et al., 2006*, and the pachycephalosaurian dinosaur *Wannanosaurus yansiensis* Hou, 1997. Two additional taxa whose exact positions within Marginocephalia remain uncertain were also included: *Micropachycephalosaurus hongtuyanensis Dong, 1978* and *Stenopelix valdensis Meyer, 1857*. These six taxa were chosen based upon their presumed basal position within Marginocephalia and because their anatomy is more completely known than other basally positioned taxa (e.g., *Chaoyangsaurus Zhao, Cheng & Xu, 1999*).

*Iguanodontia Dollo, 1888 (Sensu Sereno, 2005).* Iguanodontia is a subclade within Ornithopoda, making the inclusion of species-level exemplars from this clade crucial to elucidating the relationships of the taxa of interest in this analysis, some of which were previously proposed to be situated within Iguanodontia (e.g., *Gasparinisaura cincosaltensis*, *Talenkauen santacrucensis*). Therefore, fourteen iguanodontian species were included in this study. These species are divided into three groups. The Australian species *Muttaburrasaurus langdoni Bartholomai & Molnar, 1981*, the European species *Rhabdodon priscus Matheron, 1869*, *Zalmoxes robustus Nopcsa, 1900*, and *Zalmoxes shqiperorum Weishampel et al., 2003*, and the North American species *Tenontosaurus dossi Winkler, Murry & Jacobs, 1997* and *Tenontosaurus tilletti Ostrom, 1970* are included as non-dryomorph basal iguanodontian representatives. The European species *Callovosaurus leedsi Lydekker, 1889* and *Valdosaurus canaliculatus Galton, 1975*, the North American species *Dryosaurus altus Marsh, 1878*, and the African species *Dysalotosaurus lettowvorbecki Virchow, 1919* and *Elrhazosaurus nigeriensis Galton & Taquet, 1982* are included to represent the iguanodontian subclade Dryosauridae (sensu *Sereno, 2005*) based on the phylogenetic hypothesis published by *McDonald et al. (2010)*. Finally, the North American species *Camptosaurus dispar* (*Marsh, 1879*), the European species *Iguanodon bernissartensis Boulenger, 1881*, and the African species *Ouranosaurus nigeriensis Taquet, 1976* are included as representatives of the clade Ankylopollexia.

### Outgroup taxa

The following taxa were included as outgroups to Ornithischia. Three of these taxa were included to represent basal Saurischia, the sister taxon to Ornithischia (*Sereno, 1999*). The remaining taxa were selected based upon the phylogenetic results presented by *Nesbitt et al. (2010)* because they represent successive sister taxa to Dinosauria.

*Saurischia Seeley, 1887.* The monophyly of Dinosauria is well-supported, with Saurischia recognized as the sister taxon to Ornithischia (e.g., *Novas, 1996*; *Sereno, 1999*; *Nesbitt et al., 2009*; *Nesbitt et al., 2010*). Three basal theropod dinosaurs *Herrerasaurus ischigualastensis Reig, 1963*, *Sanjuansaurus gordilloi Alcober & Martinez, 2010*, and *Tawa hallae Nesbitt et al., 2009*, were selected to represent this clade based upon the phylogenetic results presented by *Nesbitt et al. (2009)* and *Alcober & Martinez (2010)*.

*Silesauridae Nesbitt et al., 2010.* Based upon the phylogenetic analysis by *Nesbitt et al. (2010)*, the clade Silesauridae is the sister taxon to Dinosauria, making it a preferred outgroup for analyses of basal ornithischian relationships. The two taxa selected for inclusion in this analysis, *Asilisaurus kongwe Nesbitt et al., 2010* and *Silesaurus opolensis Dzik, 2003*, represent basal and derived members of this clade, respectively.

*Marasuchus lilloensis (Romer, 1972).* This species originally was referred to the taxon *Lagosuchus (Romer, 1972)*. Subsequent revision of this taxon led *Sereno & Arcucci (1994)* to refer it to the new taxon *Marasuchus. Marasuchus liloensis* was previously included as an outgroup taxon in analyses of ornithischian relationships (e.g., *Spencer, 2007*; *Butler, Upchurch & Norman, 2008*), and the phylogenetic analysis of ornithodiran relationships by *Nesbitt et al. (2010)* confirms this species is the sister taxon to a clade composed of Silesauridae + Dinosauria. Therefore, this species was included in this analysis as a third successive outgroup to Ornithischia.

### Taxa a priori excluded from study

Several putative basal ornithischian taxa were excluded from this analysis. Many of these taxa are fragmentary and were referred to Ornithischia based upon dental characters, a practice that was recently shown to be unreliable for accurately referring fragmentary taxa to Ornithischia (e.g., *Irmis, Parker & Nesbitt, 2007*). A brief discussion of these taxa and the reasons for their exclusion is given below. It should be noted that none of the taxa discussed below were ever included in prior phylogenetic analyses of ornithischian relationships for many of the same reasons listed here.

Additionally, several taxa were named since this study was initiated and unfortunately could not be included in this version of the dataset, though certainly will be included in future versions (e.g., *Albertadromeus, Kulindadromeus, Laquintasaura, Morrosaurus, Trinisaura*; *Coria et al., 2013*; *Brown et al., 2013*; *Godefroit et al., 2014*; *Barrett et al., 2014*; *Rozadilla et al., 2016*). This latter set of taxa are not discussed below.

*Drinker nisti Bakker et al., 1990.* The holotype of *Drinker nisti* is a partial subadult individual preserving parts of the upper and lower jaws, vertebral centra, and partial fore and hind limbs (*Bakker et al., 1990*). Additional specimens referred to this taxon include isolated teeth and disarticulated postcranial elements. All of this material is from the Late Jurassic Morrison Formation of Wyoming. These specimens were briefly described and partially figured (*Bakker et al., 1990*), but their current location is unknown, preventing further elucidation of their anatomy. As a result, this taxon was excluded from the present analysis owing to a lack of relevant morphological data, despite the fact that the taxon is considered valid by some authors (e.g., *Norman et al., 2004*).

*Fulgurotherium australe Von Huene, 1932.* This poorly known taxon from the Early Cretaceous Wallangalla Sandstone Member of the Griman Creek Formation of Australia (*Agnolin et al., 2010*) is based on a partial, opalised femur. Although several other femora were referred to this taxon from this and other formations (*Rich & Rich, 1989*; *Rich & Vickers-Rich, 1999*), those referrals are suspect considering that the holotype femur does

not preserve any autapomorphic traits. Noting this problem, *Rich & Vickers-Rich (1999)* considered *F. austral* to be a "form taxon" that was useful for distinguishing between morphologically distinct subsets of femora recovered from Early Cretaceous sediments in Australia. Although *Norman et al. (2004)* considered the taxon to be valid, other authors regard it as a *nomen dubium* (e.g., *Butler, 2005*; *Agnolin et al., 2010*) and this study follows the latter opinion.

*Geranosaurus atavus Broom, 1911.* This taxon is based upon a dentary and limb elements from the Jurassic Cave Sandstone of South Africa. This taxon is currently considered to represent a *nomen dubium* (*Norman et al., 2004*).

*Gongbusaurus shiyii Dong, Zhou & Zang, 1983.* This taxon is based solely on two isolated teeth. Given the recently demonstrated difficulty of accurately assigning taxa based on isolated teeth to Ornithischia (e.g., *Irmis, Parker & Nesbitt, 2007*), this taxon is considered of dubious validity and is excluded from this study.

*Gongbusaurus wucaiwanensis Dong, 1989.* The holotype of *Gongbusaurus wucaiwanensis* consists of a fragmentary left mandible, three caudal vertebrae, and an incomplete forelimb (*Dong, 1989*). The paratype consists of two sacral vertebrae, eight caudal vertebrae, and a pair of complete hind limbs. The location of the type material of this taxon is currently unknown (*Butler, Upchurch & Norman, 2008*) and the original description is brief and poorly figured. Additional specimens were since discovered that may be referable to this taxon and remain under study by other authors (X Xu, pers. comm., 2007), but they remain unpublished. One of those specimens was personally examined by the author and it does appear to represent a distinct species, but until it is published and demonstrated that this specimen is referable to *Gongbusaurus wucaiwanensis*, it is unwise (and unethical) to include it in this analysis. Therefore, this taxon is excluded from this study.

*Hypsilophodon wielandi Galton & Jensen, 1979.* This taxon is based upon an isolated femur collected from the Early Cretaceous Lakota Sandstone of South Dakota. The specimen does not preserve any autapomorphies or a unique combination of characters and is considered to be a *nomen dubium* (*Norman et al., 2004*).

*Nanosaurus agilis Marsh, 1877b.* The hypodigm of *Nanosaurus agilis* consists of a dentary, femur, and ilium from two specimens collected from the Late Jurassic Morrison Formation of Colorado. This taxon is generally considered a *nomen dubium* owing to the lack of autapomorphic features preserved on this material (*Norman et al., 2004*); though some authors have suggested it may be diagnosably distinct (*Galton, 2007*). This study follows the former opinion and excludes *N. agilis* from the current study.

"Proctor Lake Ornithopod" (Sensu *Winkler & Murry, 1989*). This taxon is known from multiple specimens from the Early Cretaceous Twin Mountain Formation of Texas. Despite the wealth of morphological information this taxon preserves, it has yet to be formally described. It is currently under study by other researchers (D Winkler, pers. comm., 2010), precluding its inclusion in this study while that work is being completed.

*Xiaosaurus dashanpensis Dong & Tang, 1983.* *Xiaosaurus dashanpensis* is based upon a fragmentary skeleton from the Middle Jurassic lower Shaximiao Formation of Sichuan, China. As discussed by *Barrett, Butler & Knoll (2005)*, all of the apomorphies proposed by *Dong & Tang (1983)* are actually symplesiomorphies of Ornithischia, causing many to consider this taxon a *nomen dubium* (e.g., *Norman, Witmer & Weishampel, 2004a*). However, this taxon does possess a single autapomorphy of the humerus that indicates it is a valid taxon (*Barrett, Butler & Knoll, 2005*). Despite this, this study follows the advice of *Butler, Upchurch & Norman (2008)* in considering the hypodigm too fragmentary and poorly figured/described to be included in a phylogenetic analysis.

## Analysis

The data matrix was compiled using the program Mesquite v.2.74 (*Maddison & Maddison, 2009*). The final dataset was then exported as a TNT file and opened in the program Tree analysis using New Technology (TNT: *Goloboff, Farris & Nixon, 2008*). All characters were run unordered (non-additive setting in TNT). The dataset was then analyzed using the traditional search option, which is analogous to the heuristic search option in the phylogenetic program PAUP* (*Swofford, 2003*). The search was run using the tree bisection reconnection (TBR) swapping algorithm. Branches were collapsed if the minimum possible branch length was equal to zero. The search utilized 10,000 replicates with a maximum of 10,000 trees saved during each replicate. A standard bootstrap analysis was run using the program TNT for 1,000 replicates (each using a heuristic search of 100 replicates). The results are shown in Fig. 2.

## Evaluation of stratigraphic congruence

The strict consensus phylogenetic hypothesis generated by this analysis was compared to the phylogenetic hypotheses of ornithischian relationships of *Buchholz (2002)*, *Spencer (2007)* and *Butler, Upchurch & Norman (2008)* using stratigraphic consistency metrics. These metrics assume that as our understanding of the fossil record increases, phylogenetic hypotheses should become increasingly congruent with the stratigraphic record (*Pol, Norell & Siddall, 2004*). Under that assumption, the phylogenetic hypothesis that exhibits the closest fit to the fossil record best estimates the topology of the true tree. For this investigation the stratigraphic consistency measures minimum implied gap (MIG: *Benton & Storrs, 1994*; *Wills, 1999*), modified manhattan stratigraphic measure (MSM*: *Pol & Norell, 2001*) and the gap excess ratio (GER: *Wills, 1999*) were selected because those metrics are least affected by variations in tree size and shape (*Pol, Norell & Siddall, 2004*). The metric modified gap excess ratio (GER*; *Wills, Barrett & Heathcote, 2008*) was not calculated because the software used in this analysis (see below) does not provide those values. Additionally, accurately comparing stratigraphic congruence values calculated from different tree topologies requires that each tree includes an identical set of terminal taxa (*Gauthier, Kluge & Rowe, 1988*; *Wills, Barrett & Heathcote, 2008*). Therefore, when conducting these comparisons each tree topology was trimmed to include only those taxa that are present in both trees.

Calculations were conducted using the program Assistance with Stratigraphic Consistency Calculations v.4.0.0a (ASCC: *Boyd et al., 2011a*). That program provides
**Table 3   Results of the stratigraphic consistency calculations.** Values of MIG are reported in millions of years.

|  | # of taxa | MIG | GER | MSM* | Result |
|---|---|---|---|---|---|
| Full ornithischian dataset | 65 | 1908-1388 | 0.82-0.74 | 0.13-0.09 | – |
| This analysis | 19 | 480-356 | 0.80-0.67 | 0.37-0.26 | More congruent |
| *Buchholz (2002)* | 19 | 573-400 | 0.76-0.59 | 0.32-0.23 | |
| This analysis | 16 | 289-212 | 0.92-0.80 | 0.76-0.57 | Equally congruent |
| *Spencer (2007)* | 16 | 260-208 | 0.94-0.84 | 0.80-0.63 | |
| This analysis | 35 | 852-611 | 0.86-0.76 | 0.28-0.19 | More congruent |
| *Butler, Upchurch & Norman (2008)* SCC | 35 | 1582-844 | 0.77-0.49 | 0.20-0.10 | |
| This analysis | 35 | 852-611 | 0.86-0.76 | 0.28-0.19 | More congruent |
| *Butler, Upchurch & Norman (2008)* MR | 35 | 1170-877 | 0.77-0.65 | 0.19-0.14 | |
| This analysis | 28 | 723-489 | 0.86-0.75 | 0.34-0.23 | More congruent |
| *Butler, Upchurch & Norman (2008)* MAS | 28 | 816-620 | 0.81-0.70 | 0.27-0.20 | |
| This analysis | 30 | 772-531 | 0.86-0.74 | 0.32-0.21 | More congruent |
| *Butler, Upchurch & Norman (2008)* DSRC | 30 | 967-661 | 0.81-0.67 | 0.25-0.17 | |

**Notes.**

Abbreviations: DSRC, derivative strict reduced consensus tree; GER, gap excess ratio; MAS, maximum agreement subtree; MIG, minimum implied gap; MSM*, modified manhattan stratigraphic measure; MR, majority-rule consensus tree; SCC, strict component consensus tree.

the user with an interactive framework for designing an analysis and entering the required data (e.g., tree topology, taxon ages) and then calculates the final values. In situations where the tree topology being analyzed was incompletely resolved (i.e., polytomies were present), that systematic uncertainty was incorporated into the calculations using the ComPoly approach (*Boyd et al., 2011a*), which allows the full range of variation this uncertainty imparts in stratigraphic consistency values to be described. The presence of uncertainty in the age of the oldest known record for each taxon was addressed using the methods outlined by *Pol & Norell (2006)*, which allow the full range of possible dates to be defined rather than having to select a single date for each terminal taxon. Incorporating all of these methods into this analysis ensured that the conclusions drawn from comparing the resulting stratigraphic consistency values are as accurate as possible.

Six comparisons were conducted during this study. The strict consensus tree topology generated by this analysis was compared to the tree reported by *Buchholz (2002)*, the strict consensus tree by *Spencer (2007)*, and the strict consensus, majority rule consensus, maximum agreement, and derivative strict reduced consensus trees by *Butler, Upchurch & Norman (2008)*. The topology of three of these trees can be seen in Fig. 1. Values were also calculated for the unaltered strict consensus tree topology generated by this analysis (i.e., prior to being trimmed for comparison with other tree topologies). All of the resulting values are shown in Table 3.

## Reconstructing patterns of historical biogeography

Numerous researchers discussed and/or modeled patterns of historical biogeographic dispersal of ornithischian taxa (e.g., *Sereno, 1997*; *Sereno, 1999*; *Upchurch, Hunn & Norman, 2002*; *Butler et al., 2006*; *Brusatte et al., 2010*). However, patterns of biogeographic dispersal within basal Ornithischia were never reconstructed within an inclusive phylogenetic hypothesis of ornithischian relationships. Given that this study is the most comprehensive analysis of basal ornithischian relationships yet conducted, the phylogenetic hypothesis

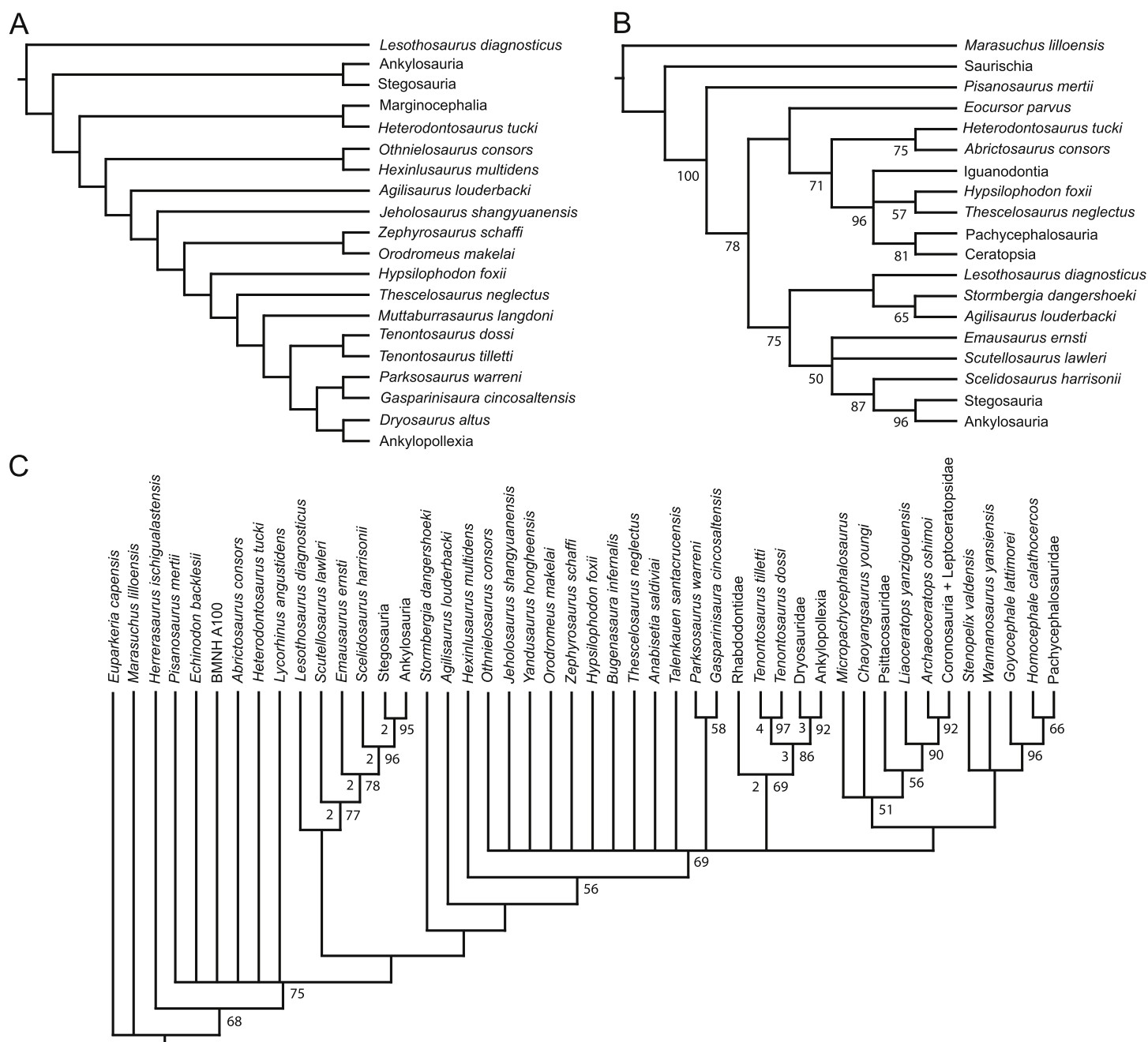

**Figure 1** **Recent phylogenetic hypotheses of basal ornithischian relationships.** Tree topology reported by *Buchholz (2002)* based on analysis of 97 characters for 20 terminal taxa (A), strict consensus of four most parsimonious trees recovered by *Spencer (2007)* based on analysis of 97 characters for 19 terminal taxa (B), and strict consensus of 756 most parsimonious trees recovered by *Butler, Upchurch & Norman (2008)* based on analysis of 221 characters for 46 terminal taxa (C). In (B) bootstrap values >50% are listed below nodes. In (C), Bremer support values >1 are to the left of nodes while bootstrap values >50% are to the right of nodes.

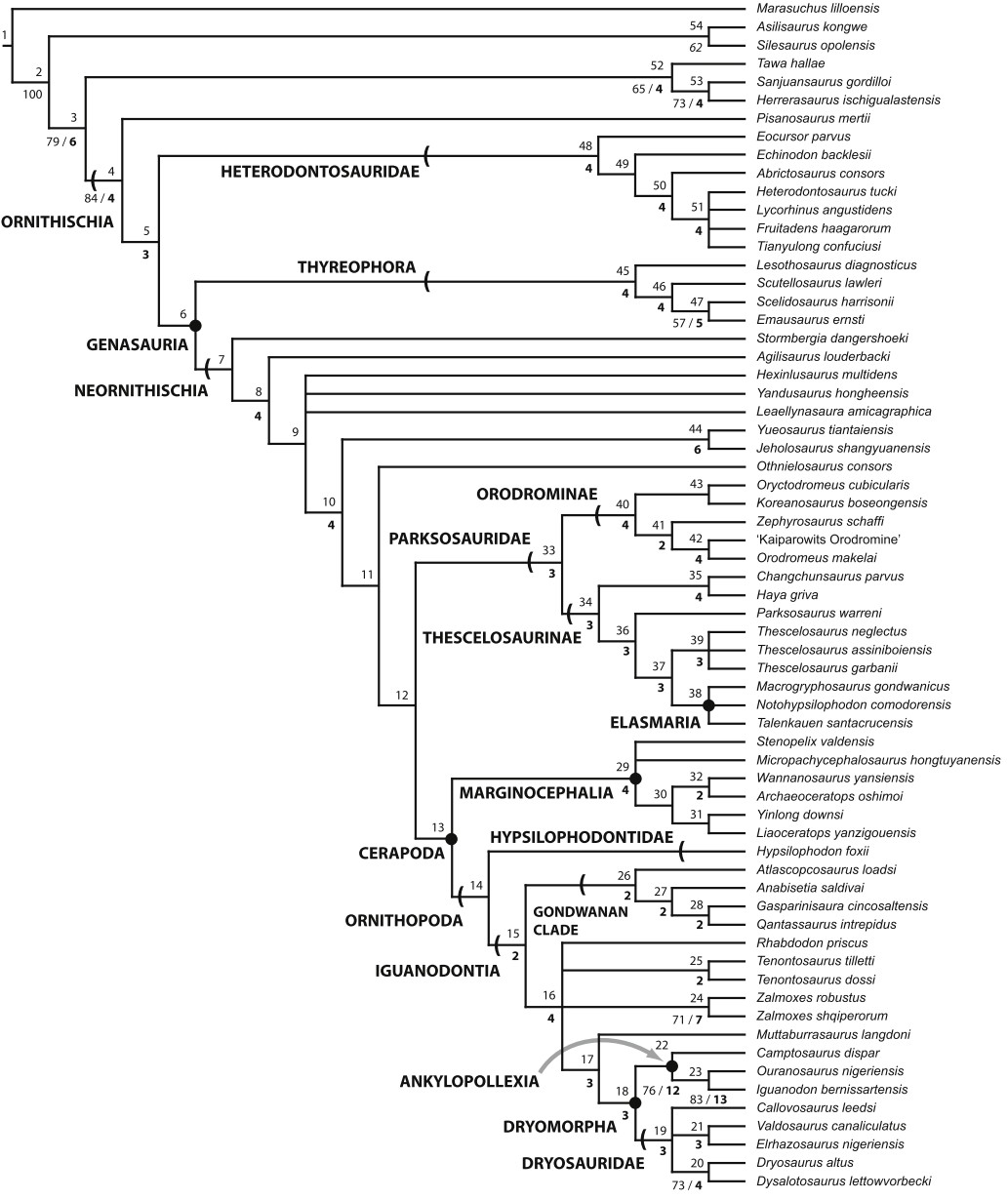

**Figure 2** **Strict consensus of the 36 most parsimonious trees recovered by this study.** Major ornithischian subclades are labeled either along branches (stem-based definitions) or at nodes (node-based definitions). See Table 1 for phylogenetic definitions. Numbers above nodes refer to the list of unambiguous character changes reported for each node in Table S4. Bold numbers beneath nodes are Bremer support numbers >1, while non-bold numbers beneath nodes are bootstrap support values >50%.

produced by this analysis provides a comprehensive framework within which to reconstruct biogeographical patterns within basal Ornithischia.

A variety of methods and programs exist for reconstructing patterns of historical biogeography (*Ronquist, 1996*; *Ronquist, 1997*; *Hausdorf, 1998*; *Ree et al., 2005*; *Ree & Smith, 2008*). The approach employed here involves incorporating time calibrated branch lengths set equal to the implied missing fossil record for each taxon when reconstructing

the geographic distribution of ancestral taxa. This allows older taxa, which are positioned closer to the ancestral nodes and are more likely to have remained in or near the ancestral geographic region, to have a larger influence over what geographic region is optimized at each node.

Reconstruction of historical biogeography was conducted using the program Mesquite v.2.74 (*Maddison & Maddison, 2009*). Three separate analyses focused on reconstructing the ancestral geographic ranges of basal ornithischian taxa were conducted. Before conducting those analyses, a new character was added to the dataset to represent the geographic range(s) of the terminal taxa. This character had six possible states, one for each continent represented in the dataset (no taxa from Antarctica were included in this analysis). Each taxon was then assigned a single state based upon their known geographic ranges. Each of the species included in this analysis are known from a single continent, precluding the need for polymorphic codings.

For all three analyses, the strict consensus topology recovered during the phylogenetic analysis was loaded into Mesquite and opened within a new tree window. In the first analysis, all branch lengths in the tree were set equal to one (Tree > Alter/Transform Branch Lengths > Assign All Branch Lengths). The trace character history option was then selected (Analysis > Trace Character History), the Stored Characters option was selected, and the Parsimony Ancestral States reconstruction method was chosen. The second analysis was similar to the first, except that in the last step the Likelihood Ancestral States reconstruction method was selected (using the default probability models).

In the third analysis, the branch lengths in the tree were manually set equal to the missing fossil record inferred for each branch (in myr). Implied missing fossil records were calculated for each branch by hand using the oldest possible age for each terminal taxon included in the analysis (see Table 2 for ages used for each taxon) and these values were assigned to their respective branches by selecting the appropriate branch in the tree and then choosing the Assign Selected Branch Lengths option (Tree > Alter/Transform Branch Lengths > Assign Selected Branch Lengths). Branch lengths for branches with no implied missing fossil record were set equal to 1. Once these data were entered, the character history of the geographic character was traced using the Likelihood Ancestral States reconstruction method (using the default probability models). The tree topology incorporating age-weighted branch lengths was not analyzed using the Parsimony Ancestral States method because that type of analysis does not take into account branch lengths when reconstructing ancestral states. The resulting character state optimizations for the geographic character were recorded for all nodes in the tree during all three analyses (see Table S5).

## RESULTS OF PHYLOGENETIC ANALYSIS

Analysis of the dataset as outlined above resulted in the recovery of thirty-six most parsimonious trees of length 868 (consistency index (CI) = 0.37; retention index (RI) = 0.65; rescaled consistency index (RCI) = 0.24). The strict consensus of these thirty-six trees and the resulting bootstrap support values are shown in Fig. 2. The details of the strict consensus tree topology are discussed in detail below. It should be noted that in the

following discussion existing phylogenetic nomenclature was utilized whenever possible (see Table 1 for a list of phylogenetic definitions). Numbers given below in parentheses refer to the character number:character state being discussed. Descriptions of the characters cited below can be found in Table S1, and a list of unambiguous character state changes within the tree is given in Table S4. All characters discussed below are unambiguously optimized synapomorphies.

## Ornithischia

The monophyly of Ornithischia is supported in this analysis by the unambiguous presence of at least a slight buccal emargination of the maxilla (19:1), the development of a distinct coronoid process of the mandible (82:1), and by the presence of a dentary contribution to the anterior portion of the coronoid process (80:1). *Pisanosaurus mertii* is recovered as the basal-most member of Ornithischia, consistent with previous analyses of the clade (e.g., *Sereno, 1999*; *Butler, Upchurch & Norman, 2008*) and is plesiomorphic with respect to all other ornithischians in possessing a lateral extension of the tibia that extends posterior to the medial margin of the fibula, but fails to contact the entire posterior margin of the fibula and calcaneum (229:1). The fragmentary nature of the holotype and only known specimen of *P. mertii* complicates optimization of several previously proposed synapomorphies of Ornithischia that cannot be assessed in this taxon. Thus, it is uncertain whether the presence of a predentary bone (2:1), an edentulous region anterior to the first premaxillary tooth (7:1), a preacetabular process of the ilium that extends anterior to the pubic peduncle (187:1), a posteroventrally oriented pubis (194:2), or a pendant fourth trochanter (219:2) represent synapomorphies of Ornithischia as a whole, or of all ornithischians excluding *Pisanosaurus*.

## Heterodontosauridae

This analysis supports the findings of *Butler, Upchurch & Norman (2008)* in placing a monophyletic Heterodontosauridae (represented by *Abrictosaurus consors*, *Echinodon becklesii*; *Eocursor parvus*, *Fruitadens haagarorum*, *Heterodontosaurus tucki*, *Lycorhinus angustidens*, and *Tianyulong confuciusi*) outside of Genasauria, contra the findings of *Sereno (1986)*, *Sereno (1999)*, *Buchholz (2002)*, *Butler (2005)*, and *Spencer (2007)*. This placement is supported by the presence of a lateral extension of the tibia that extends posterior to the entire fibula and calcaneum (229:2) in all ornithischian taxa except *Pisanosaurus*. Heterodontosauridae is placed outside of Genasauria based on the retention of a 'v-shaped' dentary symphysis (73:0) and the absence of a well-developed ventral process of the predentary (71:1). This study recovers *Eocursor parvus* as the basal-most member of Heterodontosauridae owing to the retention in this taxon of a 'V-shaped' dentary symphysis (71:1), the loss of the ventral acetabular flange of the ilium (183:1), and the presence of a horizontal brevis shelf of the ilium (189:1). Both of the latter states are present within Neornithischia in all taxa more closely related to Cerapoda than to *Agilisaurus louderbacki* (Fig. 2). *Eocursor* was previously recovered outside of Genasauria positioned between the clades Heterodontosauridae and Thyreophora (*Butler, Smith & Norman, 2007*), or as a non-cerapodan basal neornithischian (*Spencer, 2007*). While no

characters are unambiguously optimized as synapomorphies of Heterodontosauridae, all heterodontosaurids more closely related to *Heterodontosaurus* than to *Echinodon* are united in possessing maxillary and dentary teeth with denticles restricted to the apical third of the crown (134:1), as opposed to the condition in all other ornithischian taxa (except *Chaoyangsaurus youngi*) where denticles extend along the margin of most of the crown.

## Genasauria

All ornithischians except *Pisanosaurus mertii* and the heterodontosaurids are recovered within the clade Genasauria based upon the presence of a well-developed ventral process of the predentary (71:1) and a 'spout-shaped' dentary symphysis (73:1). The contents of this node-based clade (sensu *Butler, Upchurch & Norman, 2008*) are split between two less-inclusive clades, Thyreophora and Neornithischia.

## Thyreophora

The presence of postcranial osteoderms (253:1) is traditionally considered to diagnose the clade Thyreophora; however, this analysis recovers *Lesothosaurus diagnosticus*, which lacks postcranial osteoderms, as the most basal member of Thyreophora. This placement is supported based on the presence of a horizontal ridge on the surangular (86:1). The position of *L. diagnosticus* varies in recent phylogenetic analyses, with some recovering this taxon as the sister taxon to Genasauria (*Sereno, 1986*; *Sereno, 1999*; *Buchholz, 2002*), as a basal neornithischian (*Butler, 2005*), or as a basal thyreophoran (*Spencer, 2007*; *Butler, Upchurch & Norman, 2008*). The fluidity of the systematic position of *L. diagnosticus* was interpreted by *Butler, Upchurch & Norman (2008)* as evidence that the anatomy of this taxon closely resembles the basal genasaurian condition, making it a crucial taxon for evaluating the relationships of basal ornithischian taxa. Alternatively, if *L. diagnosticus* is the juvenile form of *Stormbergia dangershoeki,* as suggested by *Knoll, Padian & De Ricqles (2010)*, the retention of an unusual suite of unique and derived features in the former taxon may be an artifact of its ontogenetic status rather than a true reflection of its systematic position. All thyreophorans to the exclusion of *L. diagnosticus* are united in possessing postcranial osteoderms (253:1) and an anterior process of the jugal that is mediolaterally broader than dorsoventrally deep (32:1). *Scelidosaurus harrisonii* and *Emausaurus ernsti* are united in possessing the apomorphic condition of a dentary tooth row that in sinuous in lateral view (78:1).

## Neornithischia

Neornithischian taxa are united in possessing a tab shaped obturator process on the ischium (203:1) and an articulation between a sacral rib and the ischiadic peduncle of the ilium (190:1). The former character is lost in Marginocephalia and Rhabdodontidae. Like *Butler, Upchurch & Norman (2008)* and *Spencer (2007)*, this analysis places several taxa outside of the node-based clade Cerapoda as non-cerapodan basal neornithischians, though the set of taxa here included under this designation is larger than in any previous analysis. Twenty-two taxa in this study are recovered as non-cerapodan basal neornithischians (Fig. 2).

*Stormbergia dangershoeki* is placed below *Agilisaurus louderbacki* as the most-basal neornithischian taxon based upon the retention in *S. dangershoeki* of a pubic peduncle of the ilium that is larger than the ischiadic peduncle (192:0). Complementarily, the presence of a reduced pubic peduncle (192:1) is an unambiguously optimized synapomorphy of all neornithischian taxa more closely related to Cerapoda than to *S. dangershoeki*. All neornithischian taxa more closely related to Cerapoda than to *Agilisaurus louderbacki* lack a ventral acetabular flange of the ilium (183:1; state present convergently in heterodontosaurids), possess a weakly developed or absent supra-acetabular rim on the ilium (184:1; reversed in *Zalmoxes* and present convergently in *L. diagnosticus*), and display a horizontal brevis shelf on the ilium (189:1; present convergently in heterodontosaurids).

*Jeholosaurus shangyuanensis* and *Yueosaurus tiantaiensis* both possess a relatively straight humerus that lacks a posterior flexure at the level of the deltopectoral crest (167:0), which is plesiomorphic for Ornithischia. The clade consisting of *J. shangyuanensis* + *Y. tiantaiensis* is unambiguously united with all other neornithischians in possessing a distinct 'trench' (i.e., *fossa trochanteris*) between the greater trochanter and the head of the femur (212:1). Other important characters that unite these taxa include the presence of six or more sacral vertebrae (148:2; convergently present in some heterodontosaurids), lateral swelling of the ischiadic peduncle of the ilium (191:1; apomorphically reversed in *Stenopelix valdensis* (*Butler & Sullivan, 2009*)), and a lesser trochanter of the femur that is anteroposteriorly narrow and closely appressed to the greater trochanter (217:2; also present in *Leaellynasaura amicagraphica* and some heterodontosaurids and reversed in *Callovosaurus leedsi*) with its dorsal extent approximately level with the head of the femur (218:1; also present in some heterodontosaurids).

*Othnielosaurus consors* is positioned as more closely related to Cerapoda than to the clade consisting of *J. shangyuanensis* + *Y. tiantaiensis* based on the presence of neural spines on the caudal vertebrae that extend posteriorly beyond the caudal centra (152:1; reversed in *Orodromeus makelai*, *Parksosaurus warreni* and *Zalmoxes robustus*) and a tibia with a triangular cross-sectional shape (227:0; reversed in *Koreanosaurus boseongensis*, *Parksosaurus warreni*, and some iguanodontians, convergently present in *Stormbergia dangershoeki*). *Othnielosaurus consors* is placed below a clade consisting of Cerapoda + Parksosauridae (see Table 1 for definitions) based on the retention of a dentary with a dorsoventral height that is less than 20% of its length (77:0) and dentary teeth that lack a prominent primary ridge (139:0), though both of these characters display a relatively low CI (0.25 and 0.17, respectively).

## Cerapoda

Cerapodan taxa differ from all other neornithischian taxa in possessing dorsomedially sloped or horizontal distal condyles of the quadrate (52:0; convergently present in Thyreophora), maxillary crowns that taper to the root (120:1; convergently present in *Jeholosaurus shangyuanensis* and *Heterodontosaurus tucki*, and reversed in *Yinlong downsi*), asymmetrically distributed enamel on the 'cheek' teeth (123:1; convergently present in *Abrictosaurus consors* and *Heterodontosaurus tucki*), and dentary crowns with ridges restricted to the lingual surface (124:1; reversed in the unnamed clade of Gondwanan

iguanodontians recovered in this analysis and convergently present in *Heterodontosaurus tucki*). This node-based clade (sensu *Butler, Upchurch & Norman, 2008*) is subdivided into the stem-based clades Marginocephalia and Ornithopoda (see Table 1 for definitions).

## Marginocephalia

A monophyletic Marginocephalia is recovered; however, because the cranial morphology of *Stenopelix valdensis* and *Micropachycephalosaurus hongtuyanensis* remain poorly understood, the presence of both a parietosquamosal shelf (58:1) and exclusion of the premaxillae from the internal nares (11:1) are not unambiguously optimized as synapomorphies of this clade. However, the presence of a dorsoventrally flattened anterior process of the pubis (197:3) is recovered as an unambiguous synapomorphy of the clade. Another proposed marginocephalian synapomorphy, a shortened postpubic process (*Butler, Upchurch & Norman, 2008*), was not assessed in this study because it could only be scored for a single marginocephalian taxon (*Stenopelix valdensis*) included in this study (published data for *Yinlong downsi* does not sufficiently describe the anatomy of the pubis).

## Ornithopoda

This study recovers a more restricted Ornithopoda than previously proposed (Fig. 2), consisting only of *Hypsilophodon foxii* + Iguanodontia. The stem-based clade Ornithopoda and the recently named node-based clade Clypeodonta (*Hypsilophodon foxii* + *Edmontosaurus regalis*: *Norman, 2015*) have the same taxonomic contents in this study; therefore, the latter clade is not discussed herein because Ornithopoda has priority. Ornithopod taxa are unambiguously united in possessing maxillary crowns that are shorter than wide (132: states 0 or 1; reversed in *Qantassaurus intrepidus* and convergently present in *Pisanosaurus mertii*).

## Hypsilophodontidae

The only member of this clade is *Hypsilophodon foxii*, supporting prior assertions that the traditional contents of this clade represent a paraphyletic assemblage of taxa and not a monophyletic grouping (e.g., *Scheetz, 1999*; *Buchholz, 2002*; *Butler, Upchurch & Norman, 2008*), contra the findings of *Sereno (1986)*, *Sereno (1999)*, *Butler (2005)* and *Spencer (2007)*.

## Iguanodontia

Iguanodontians are unambiguously united in possessing a jugal wing of the quadrate that is positioned well dorsal to the distal condyles (49:1). They also display sacral neural spines that are at least twice the height of the sacral centra (149:1 or 2; state 1 convergently acquired in *Oryctodromeus cubicularis*). Four of the taxa of interest to this study are positioned at the base of Iguanodontia as part of a previously unrecognized clade: the South American taxa *Anabisetia saldiviai* and *Gasparinisaura cincosaltensis*, and the Australian taxa *Atlascopcosaurus loadsi* and *Qantassaurus intrepidus*. The placement of *G. cincosaltensis* in a clade at the base of Iguanodontia makes the contents of the stem-based Iguanodontia (sensu *Sereno, 2005*) and the node-based Euiguanodontia (sensu *Coria & Salgado, 1996*) identical. As Iguanodontia has priority, the clade name Euiguanodontia is not used in the remainder of this discussion.

## Unnamed clade of Gondwanan taxa

Character support is low for this clade, which is expected given that the Australian taxa are known only from disarticulated maxillae and dentaries. They are united in possessing mandibular teeth with vertical ridges present on both sides of the crown (124:0), which is the plesiomorphic ornithischian condition. These taxa are positioned basal to all other iguanodontians because they lack 'lozenge-shaped' dentary crowns (136:0), and robust (i.e., thickened) postorbitals (61:1), which are unambiguously optimized synapomorphies of more derived iguanodontians. The recovery of this clade supports hypotheses presented by various authors that some Australian (e.g., *Atlascopcosaurus*) and South American taxa (e.g., *Gasparinisaura*) were closely related (e.g., *Norman et al., 2004*; *Agnolin et al., 2010*). The characters supporting the remainder of the iguanodontian taxa included in this study are not discussed as they were not a part of the taxa of interest.

## Parksosauridae

This study supports the results of *Boyd et al. (2009)*, *Brown & Druckenmiller (2011)* and *Brown et al. (2013)* in recovering a clade of taxa traditionally recognized as 'hypsilophodontid' taxa; however, *Hypsilophodon foxii* is not placed within this clade, precluding the application of the clade name Hypsilophodontidae. Several other names could be applied to this clade. The stem-based clade Parksosauridae was first defined by *Buchholz (2002)* as the most inclusive clade containing *Parksosaurus warreni*, but not *Hypsilophodon foxii*, *Dryosaurus altus*, or *Iguanodon bernissartensis*. Alternatively, *Brown et al. (2013)* defined the node-based clade Thescelosauridae as *Thescelosaurus neglectus*, *Orodromeus makelai*, their most recent common ancestor, and all of its descendants. In this study both of those clade names contain the exact same set of fourteen taxa. Given that the former name has taxonomic priority over the latter name, this study utilizes the name Parksosauridae (Fig. 2), though the definition is slightly modified to ensure stability (see Table 1).

In this study all parksosaurids are unambiguously united in possessing a posterolateral concavity within the posterior end of the premaxilla, near the lateral margin, for receipt of the anterolateral boss of the maxilla (14:1). They also possess a modestly flared oral margin of the premaxilla (5:1; reversed in *Haya griva* and *Orodromeus makelai* and convergently present in *Agilisaurus louderbacki*), fused premaxillae (255:1; reversed in *Haya griva* and *Orodromeus makelai*), a flattened lateral surface of the greater trochanter (213:1; convergently evolved in *Gasparinisaura cincosaltensis* and *Zalmoxes*), and a braincase with an angle of less than 35° between its base and the long axis (98:1; convergently present in *Hypsilophodon foxii*).

## Orodrominae

The contents of Orodrominae recovered in this study match that proposed in *Brown et al. (2013)*, consisting of four North American taxa (*Orodromeus makelai*, *Oryctodromeus cubicularis*, *Zephyrosaurus schaffi*, and the 'Kaiparowits orodromine') and one Asian taxon (*Koreanosaurus boseongensis*). Two unambiguous synapomorphies unite these taxa: presence of a sharp and pronounced scapular spine (158:1); and, fibular shaft

'D-shaped' in cross-section throughout it length (233:1). Additionally, these taxa are united by the presence of a sharp ventral keel on the cervical vertebrae (143:1; convergently evolved in the dryosaurid *Valdosaurus canaliculatus*). Among the five taxa here referred to Orodrominae, *O. makelai*, *Z. schaffi*, and the 'Kaiparowits orodromine' are united in displaying a tall, posterolaterally directed jugal horn (38:3; convergently present in *Heterodontosaurus tucki*).

### Thescelosaurinae

The clade name Thescelosaurinae was first proposed by *Sternberg (1940)* and was first phylogenetically defined by *Brown & Druckenmiller (2011)*, though a slightly different definition is used here to ensure stability (see Table 1). The stem-based clade Thescelosaurinae is the sister-taxon to the stem-based clade Orodrominae, which together comprise the stem-based clade Parksosauridae. Thescelosaurines are united in possessing two supraorbital bones that are not fused to the orbital margin (23:2; convergently present in *Agilisaurus louderbacki*) and a dorsally projecting 'finger-like' process on the surangular anterior to the jaw joint (86:2; convergently present in the iguanodontian *Tenontosaurus tilletti*), though both of these characters suffer from missing data both within and outside the clade. *Haya griva* and *Changchunsaurus parvus* differ from all other thescelosaurines in possessing dentaries with parallel dorsal and ventral margins (75:1) and an anterior tip of the dentary positioned at approximately mid-height (74:1). The remaining seven thescelosaurines differ from most basal ornithischians in possessing a femur with the fourth trochanter extending onto the distal half of the shaft (221:1; convergently present in some iguanodontian taxa and other large-bodied ornithischian taxa not included in this analysis), partial ossification of the sternal segments of the cranial dorsal ribs (157:1; convergently present in *Othnielosaurus consors* and *Hypsilophodon foxii*), placement of the obturator process of the ischium along the distal 60% of the ischial shaft (204:1; convergently present in *Hypsilophodon foxii*), and a femoral shaft bowed in anterior view (209:1; convergently present in *Jeholosaurus shangyuanensis* and some basal iguanodontians). The three species of *Thescelosaurus* and Elasmaria are recovered as sister taxa to the exclusion of all other parksosaurids based on the presence of an ilium with a sinuous dorsal margin (185:1; straight in all other parksosaurids), a low olecranon process of the ulna (169:0), and the presence of a femur that is longer than the tibia (226:1; convergently present in most basal iguanodontians and *Scelidosaurus harrisonii*).

### Elasmaria

This analysis recovers a slightly more inclusive Elasmaria clade than was originally proposed (see *Calvo, Porfiri & Novas, 2007* and Table 1), though not as inclusive as in other recent analyses (*Rozadilla et al., 2016*). In addition to *Talenkauen santacrucensis* and *Macrogryphosaurus gondwanicus*, the Patagonian taxon *Notohypsilophodon comodorensis* is placed within this clade. All three taxa possess a highly reduced deltopectoral crest on the humerus (168:2; convergently present in basal iguanodontian *Anabisetia saldiviai*). *T. santacrucensis* and *M. gondwanicus* both retain the primitive condition of an epiphysis on cervical vertebra three (145:0; present in all ornithischians positioned below *Agilisaurus*

*louderbacki*), which *Calvo, Porfiri & Novas (2007)* used to diagnose this clade, and an ovoid, or subcylindrical, ischial shaft (205:1; convergently present in *Zephyrosaurus schaffi* and many iguanodontians). Because the presence or absence of both of these characters cannot be assessed in *Notohypsilophodon* due to preservational issues, it is unclear if they unite all elasmarians, or if they are diagnostic for a more restricted clade composed of *T. santacrucensis* and *M. gondwanicus*. The presence of thin mineralized plates on the anterior portion of the thoracic ribcage (=intercostal plates: *Butler & Galton, 2008*; *Boyd, Cleland & Novas, 2011*) was also proposed to diagnose this clade (e.g., *Calvo, Porfiri & Novas, 2007*; *Rozadilla et al., 2016*). However, these structures are more widely distributed among basal neornithischian and basal ornithopod taxa (i.e., *Hypsilophodon foxii, Othnielosaurus consors, Parksosaurus warreni,* and *Thescelosaurus neglectus*: (*Butler & Galton, 2008*; *Boyd, Cleland & Novas, 2011b*) and do not diagnose this clade.

## HISTORICAL BIOGEOGRAPHY OF ORNITHISCHIA

The results of the three analyses of basal ornithischian historical biogeography are presented in Fig. 3 (parsimony) and Fig. 4 (likelihood). To simplify comparisons between these sets of results, they will be referred to as follows: parsimony-based analysis (PB); likelihood-based analysis with equal branch lengths (LEB); and, likelihood-based analysis with time calibrated branch lengths set equal to implied missing fossil records (LFR). Additionally, a time-calibrated version of the strict consensus tree presented from Fig. 2 is shown in Fig. 5. By combining the temporal and geographic information presented in these figures, the biogeographic history of basal Ornithischia can be reconstructed and discussed in detail.

The LFR analysis reconstructs the ancestral area of the common ancestor of Dinosauria + Silesauridae (sensu *Nesbitt et al., 2010*) as Africa, while the other two analyses place the origin in South America. This difference reflects the fact that the African taxon *Asilisaurus kongwe Nesbitt et al., 2010* is the oldest taxon included in this analysis, giving it more weight in the LFR analysis. All three analyses reconstruct the origin of Dinosauria and Ornithischia in South America, which is consistent with some prior proposals (e.g., *Sereno, 1997*). Considering two of the three basal theropods included in this study and the basal-most ornithischian taxon, *Pisanosaurus mertii*, are all from South America, this result is unsurprising. Ornithischia diverged from its sister taxon Saurischia by the early Late Triassic at the very latest (Fig. 5).

The ancestral area of the most recent common ancestor of the clade consisting of Heterodontosauridae + Genasauria is optimized as Africa by all three analyses, with this split likely taking place during the Late Triassic. Likewise, all three analyses reconstruct a period of rapid diversification of Heterodontosauridae to have occurred in Africa during either the Late Triassic or Early Jurassic (contra the results of *Pol, Rauhut & Becerra (2011)*), with the lineages leading to *Echinodon becklesii, Fruitadens haagarorum* and *Tianyulong confuciusi* later dispersing into Europe, North America, and Asia, respectively. These dispersals could have occurred anytime during the Jurassic.

The origin of Genasauria is hypothesized by all three analyses to have occurred in Africa during the Early Jurassic at the latest, and possibly during the Late Triassic, and the

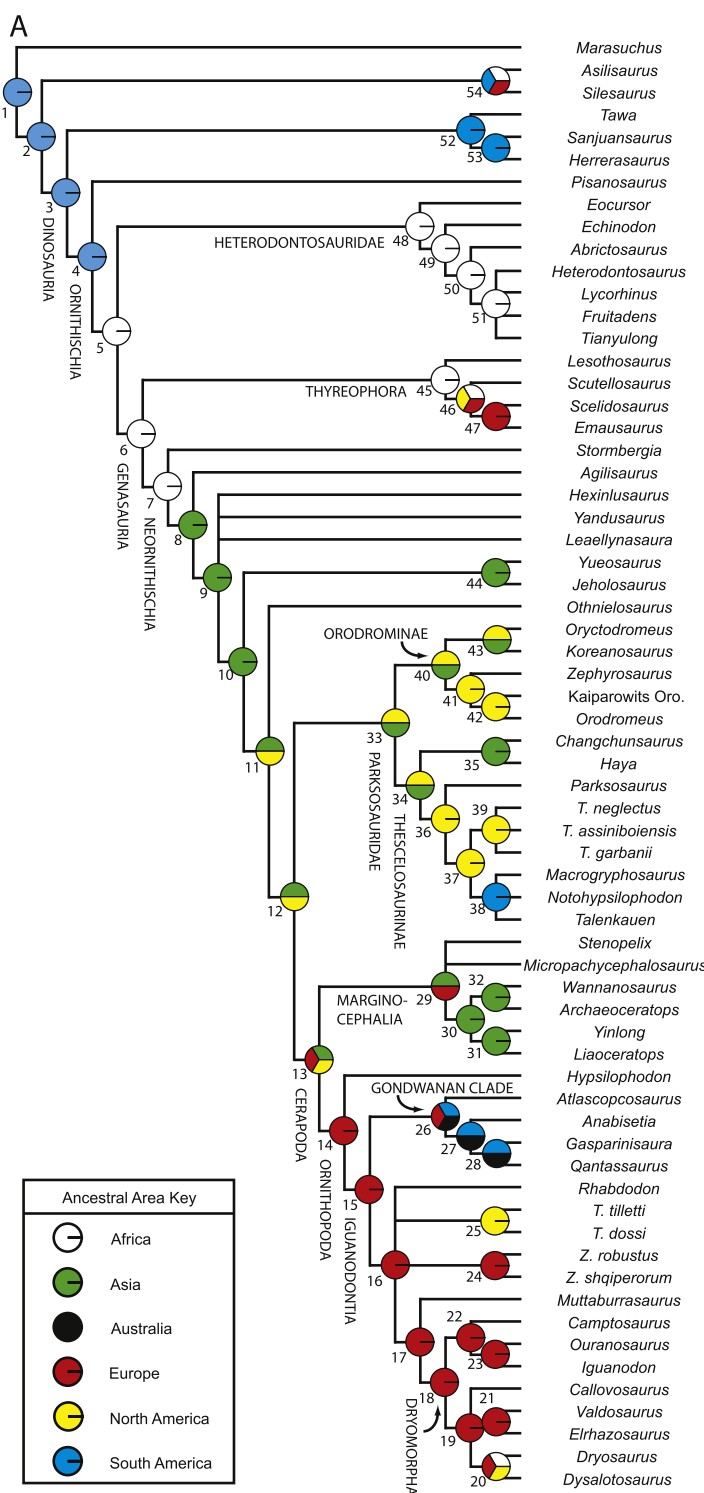

**Figure 3 Parsimony-based reconstructions of ancestral geographic areas.** Tree topology based on Fig. 2. The pie charts at each node represent the level of support for each ancestral (see Table S5 for precise values). Each color represents a different geographic area (see key). Numbers next to nodes refer to those used in Table S5.

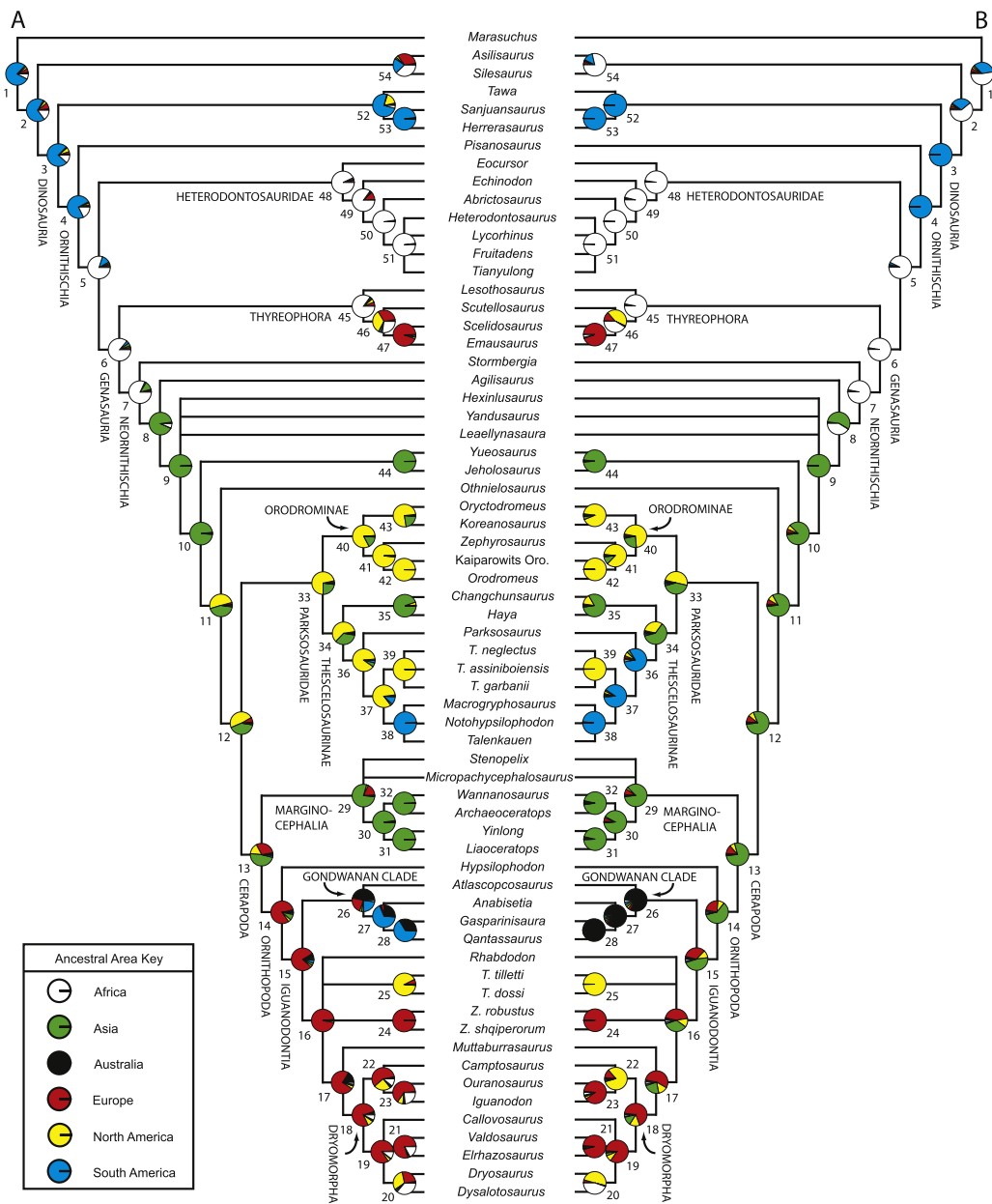

**Figure 4** **Likelihood-based reconstructions of ancestral geographic areas.** Results obtained when all branch lengths were equal (A) versus results obtained when time calibrated branch lengths were included and set equal to inferred missing fossil records (B). Tree topology based on Fig. 2. The pie charts at each node represent the level of support for each ancestral area (See Table S5 for values). Each color represents a different geographic area (see key). Numbers next to nodes refer to Table S5.

early diversification of Thyreophora also transpired in Africa, assuming the placement of *Lesothosaurus diagnosticus* at the base of this clade is accurate. There exists disagreement regarding the pattern of dispersal within Thyreophora. The LFR and LEB analyses slightly favor a scenario where basal thyreophorans dispersed from Africa into North America (giving rise to the *Scutellosaurus lawleri* lineage) and then migrating into Europe. The

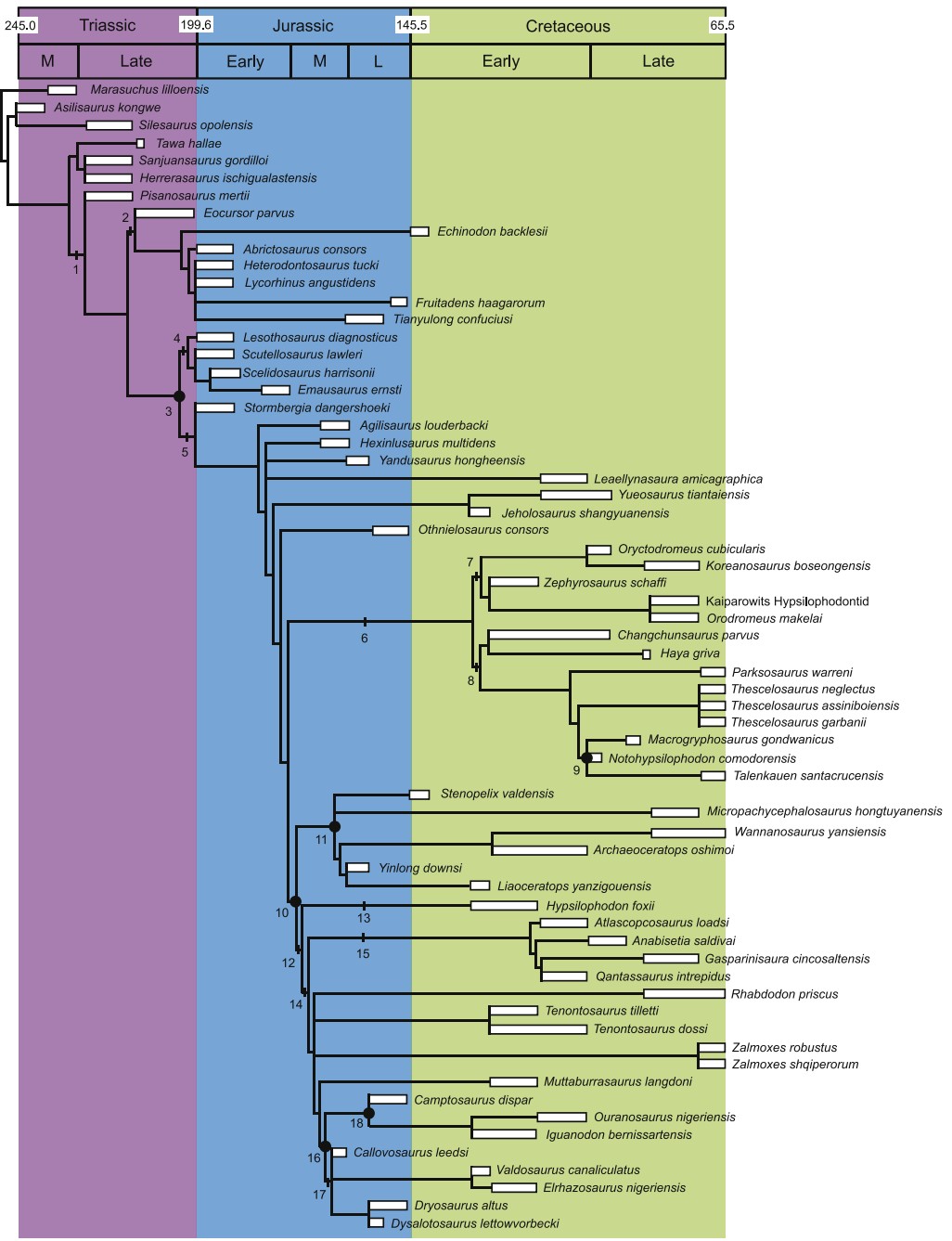

**Figure 5** **Time-calibrated phylogeny of Ornithischia.** White boxes indicate the uncertainty around the age of first appearance for each terminal taxon (not the known occurrences), while black lines represent implied missing fossil records (i.e., ghost lineages). Note: some branches are necessarily drawn deeper in time due to drawing constraints. Numbers positioned along branches or at nodes indicate the position of major ornithischian subclades. 1, Ornithischia; 2, Heterodontosauridae; 3, Genasauria; 4, Thyreophora; 5, Neornithischia; 6, Parksosauridae; 7, Orodrominae; 8, Thescelosaurinae; 9, Elasmaria; 10, Cerapoda; 11, Marginocephalia; 12, Ornithopoda; 13, Hypsilophodontidae; 14, Iguanodontia; 15, unnamed Gondwanan clade; 16, Dryomorpha; 17, Dryosauridae; 18, Ankylopollexia.

PB analysis is equivocal as to whether this is the case or if basal thyreophorans dispersed from Africa directly into Europe, with *Scutellosaurus lawleri* dispersing separately into North America. Either way, the diversification of these basal members of Thyreophora was completed before the late Early Jurassic. However, if *L. diagnosticus* is not a basal thyreophoran, then these reconstructed patterns could change dramatically.

The species *Stormbergia dangershoeki* constrains the origin of the Neornithischia to the Early Jurassic at the latest, and all three analyses agree that this clade arose in Africa. Sometime before the late Middle Jurassic there is an extensive radiation of neornithischian taxa, though the poor fossil record of neornithischian taxa during the Early and early Middle Jurassic make it impossible to determine precisely how rapidly this radiation occurred. However, all three biogeographic analyses agree that this radiation occurred in Asia (Figs. 3 and 4). All three analyses remain in agreement regarding the diversification of Neornithischia occurring within Asia until the most recent common ancestor of the clade consisting of *Othnielosaurus consors* + (Parksosauridae + Cerapoda). At this node, the LEB analysis slightly favors North America as the ancestral area (50.7% versus 42.9% for Asia), while the LFR analysis strongly favors Asia (81.1% versus 8.8% for North America). The PB analysis is equivocal. The situation is similar for the most recent common ancestor of the clade consisting of Parksosauridae + Cerapoda.

The earliest known parksosaurid taxa, *Changchunsaurus parvus* and *Zephyrosaurus schaffi*, are present in the middle Early Cretaceous (Fig. 5). However, a long ghost lineage is present for Parksosauridae, stretching from at least the Bathonian until the Aptian, a time span of at least 40 myr. The LEB and LFR analyses agree that the basal split within Parksosauridae that gave rise to the clades Orodrominae and Thescelosaurinae occurred in North America by the Aptian (the PB analysis is undecided between North America and Asia). Both the LEB and LFR analyses also agree that the diversification of orodromine taxa occurred in North America during the Cretaceous, with the lineage leading to *Koreanosaurus boseongensis* splitting from *Oryctodromeus cubicularis* either during or prior to the Cenomanian, with the former taxon eventually dispersing into Asia by the Santonian (Figs. 4 and 5). The PB analysis largely agrees with this interpretation, though it is equivocal as to whether at least some of the diversification of orodromine taxa occurred in Asia (Fig. 3).

Substantial disagreement exists between all three analyses regarding the pattern of geographic dispersals present within Thescelosaurinae. In the PB analysis (Fig. 3), thescelosaurines originated in either North America or Asia. The most recent common ancestor of the clade composed of *Changchunsaurus parvus* and *Haya griva* was located in Asia, and this clade arose by the Aptian. The ancestral area for most of the remaining thescelosaurines was North America, though a single lineage dispersed to South America by the Cenomanian, giving rise to Elasmaria. The LEB analysis largely agrees with this interpretation (Fig. 4A), though it sets the origin of Thescelosaurinae within North America, with the clade consisting of *C. parvus* and *H. griva* dispersing into Asia by the Aptian. The results of the LRF analysis strongly contrast with both of the other analyses. The LRF analysis (Fig. 4B) places the basal split within Thescelosaurinae in Asia prior to the Aptian. The sister taxon to the *C. parvus* + *H. griva* clade then migrates into South America (possibly

by way of North America). Prior to the Cenomanian, two thescelosaurine lineages disperse into North America from South America. The first gives rise to *Parksosaurus warreni*, while the second gives rise to the *Thescelosaurus* clade.

The LFR and LEB analyses place the origin of Cerapoda within Asia prior to the late Middle Jurassic (Fig. 4), though the PB analysis finds North America, Europe, and Asia equally likely areas (Fig. 3). The diversification of Marginocephalia occurred most likely in Asia by the Late Jurassic (the PB analysis is uncertain if this occurs in Asia or Europe owing to the basal position of *Stenopelix valdensis*).

Extensive disagreement exists between each of the three analyses concerning the biogeographic history of basal iguanodontians, and each set of results will be discussed separately. The PB analysis (Fig. 3) places the origin of Iguanodontia in Europe. The ancestral location of the newly recognized Gondwanan clade of iguanodontians was either in Europe, Australia, or South America. The pattern of geographic dispersals within the Gondwanan iguanodontian clade is not sufficiently resolved in this analysis to permit further comment. The majority of the remaining iguanodontian taxa were endemic to Europe, with a single lineage dispersing to North America by the Aptian that gave rise to *Tenontosaurus*. The pattern of geographic dispersals involving *Dryosaurus altus* and *Dysalotosaurus lettowvorbecki* are unresolved. The LEB analysis (Fig. 4A) also places the origin of Iguanodontia in Europe; however, the basal split within the Gondwanan iguanodontian clade is placed in Australia. After this split, the ancestral area of the remaining members of the clade moves to South America, with *Qantassaurus intrepidus* dispersing back to Australia before the Valanginian. The remaining diversification follows that recovered by the PB analysis, though it recovers Africa as the ancestral area for the most recent common ancestor of *D. altus* and *D. lettowvorbecki*, with the former migrating into North America by the Late Jurassic.

The results of the LRF analysis contrast sharply with those of both the PB and LEB analyses (Fig. 4B). The LRF analysis places the origin of the Iguanodontia in Asia, requiring the lineage leading to *Hypsilophodon foxii* to disperse into Europe by the Aptian. The most recent common ancestor of the Gondwanan iguanodontian clade and Dryomorpha was also situated in Asia. The basal divergence within the Gondwanan iguanodontian clade occurred within Australia, as did all subsequent diversification within the clade, requiring two separate dispersals from Australia into South America. Above this clade, the ancestral area changes to Europe, with the lineage leading to the *Tenontosaurus* clade later migrating to North America and diversifying. The LFR analysis contradicts the LEB analysis in that the most recent common ancestor of the clade containing *D. altus* and *D. lettowvorbecki* migrates from Europe into North America, with the latter taxon then migrating to Africa (Fig. 4B versus Fig. 4A). One additional difference between the LFR analysis and the others is that the origin of Ankylopollexia is hypothesized to have occurred in North America and not Europe owing to the basal placement of the Jurassic taxon *Camptosaurus dispar*.

## RESULTS OF STRATIGRAPHIC CONGRUENCE ANALYSIS

The results of the stratigraphic congruence analysis are shown in Table 3. Comparisons were limited in some cases by the necessity of trimming each tree topology to only
include congruent sets of taxa. In the most extreme case, the strict consensus topology generated by this analysis was trimmed from sixty-five terminal taxa to sixteen to facilitate comparison with the strict consensus topology from *Spencer (2007)*, limiting the amount of data available to compare these tree topologies (see Fig. 1B versus Fig. 2). An additional complicating factor was the high number of taxa placed within polytomies in each tree topology (e.g., 14 out of 35 taxa are placed in unresolved positions in the strict consensus tree of *Butler, Upchurch & Norman, 2008*). As a result, the minimum and maximum recovered values for each metric tend to be are highly disparate, lowering the chances of being able to select one tree topology as more congruent with the stratigraphic record of first appearances than another.

Despite these methodological difficulties, most of these comparisons resulted in the selection of one topology as more congruent with the stratigraphic record (Table 3). In five of the six comparisons made, the strict consensus tree produced by this analysis was found to be more stratigraphically congruent than the alternative topology, and in the sixth case the two trees were found to be equally congruent (Table 3). This latter result may be at least in part due to the small number of taxa shared between these two analyses (sixteen shared taxa); however, the topology from *Buchholz (2002)* only shares nineteen taxa in common with the strict consensus topology of this analysis, and in that case the topology from this analysis is clearly more congruent with the stratigraphic record of first appearances (Table 3). Most importantly, the strict consensus tree topology recovered in this study is found to be more congruent with the stratigraphic record of first appearances than any of the tree topologies put forth by *Butler, Upchurch & Norman (2008)*, which was the most comprehensive analysis of basal ornithischian relationships prior to this study.

## DISCUSSION

The strict consensus topology produced by this analysis (Fig. 2) is the most inclusive and well-resolved phylogenetic hypothesis of ornithischian relationships to date. This tree topology is equally congruent or more congruent with the stratigraphic record of first appearances than any other ornithischian phylogeny published in the last decade and a half (Fig. 5; Table 3). Comparing the results of this study to those of other recently published ornithischian phylogenetic hypotheses (i.e., *Buchholz, 2002*; *Spencer, 2007*; and *Butler, Upchurch & Norman, 2008*) provides important insights into those areas of the ornithischian evolutionary tree where our understanding is improving, where a consensus is beginning to be reached on contentious relationships, and where further improvement is needed.

Both this analysis and that of *Butler, Upchurch & Norman (2008)* recover a monophyletic Heterodontosauridae positioned outside of Genasauria at the base of Ornithischia, though more derived than *Pisanosaurus mertii*. This is in strong contrast to the traditional placement of Heterodontosauridae within Ornithopoda, a placement that has not been recovered since *Sereno (1999)*. Thus, support is building for the removal of Heterodontosauridae from Genasauria. However, heterodontosaurids do convergently share some features with basal neornithischian taxa more closely related to Cerapoda

than to *Agilisaurus louderbacki* (e.g., loss of a ventral acetabular flange on the ilium). This may account for the recovery of Heterodontosauridae at the base of Neornithischia outside of Cerapoda by the more restricted analysis conducted by *Spencer (2007)*. *Buchholz (2002)* included only one heterodontosaurid, *Heterodontosaurus tucki,* in his analysis of ornithischian relationships, recovering it as the sister taxon to a supraspecific terminal taxon representing Marginocephalia. The only other phylogenetic analysis to recover this set of relationships also included *H. tucki* as the only representative of Heterodontosauridae (*Xu et al., 2006*). These unconventional results are likely a result of the fact that *H. tucki* is a relatively derived member of Heterodontosauridae (*Pol, Rauhut & Becerra, 2011*; this analysis), and is not an ideal exemplar species for representing Heterodontosauridae in phylogenetic analyses, at least not by itself.

This analysis recovers *Eocursor parvus* as a non-genasaurian ornithischian, as did *Butler, Smith & Norman (2007)* and *Pol, Rauhut & Becerra (2011)*, contrasting with its placement as a basal neornithischian by *Spencer (2007)*. Unlike *Butler, Smith & Norman (2007)* and *Pol, Rauhut & Becerra (2011)*, this analysis identifies *E. parvus* as the basal-most heterodontosaurid. *Butler (2010)* provides a detailed list of features that separate *E. parvus* from heterodontosaurids, many of which are included as characters in this analysis (e.g., distribution of denticles on the tooth crowns and development of the coronoid process). Despite the inclusion of this evidence, *Eocursor* is positioned at the base of Heterodontosauridae based in part on the presence of some of the same features that are convergently shared between other heterodontosaurids and basal neornithischians more closely related to Cerapoda than to *Agilisaurus louderbacki* (e.g., presence of a ventral acetabular flange on the ilium). As such, it seems more plausible that *Eocursor* was a basal heterodontosaurid, which requires only two losses of these features within Ornithischia, as opposed to interpreting three independent losses near the base of Ornithischia.

This analysis recovers a very restricted Ornithopoda, which contains only *Hypsilophodon foxii* and Iguanodontia. Such a restricted Ornithopoda has never been recovered before in a published analysis, though the largely unpublished analysis of ornithischian relationships summarized in *Liu (2004)* recovered an even more restricted Ornithopoda that included an identical set of taxa as Iguanodontia. The reduced size of Ornithopoda in the study presented here is a result of the relatively high placement of Marginocephalia on the tree relative to other analyses (e.g., *Sereno, 1999*; *Butler, Upchurch & Norman, 2008*). As a result, most taxa previously referred to the Hypsilophodontidae are now non-cerapodan basal neornithischians, with the exception of *H. foxii*. Despite not being strongly supported in the bootstrap analysis (Fig. 2), the placement of Marginocephalia on the tree is by far the most parsimonious placement given the character data analyzed. Moving Marginocephalia down the tree a single node to a position below Parksosauridae adds seven steps to the total tree length. Positioning Marginocephalia further down below *Jeholosaurus shangyuanensis* (the location recovered by *Butler, Upchurch & Norman (2008)*) increases the tree length by nine steps. Thus, the recovered position of Marginocephalia is relatively well supported by the character data used in this study.

A clade composed solely of North American basal neornithischians was first recovered by *Boyd et al. (2009)* in their analysis of specimens previously referred to *Thescelosaurus*. This
analysis recovers a similar clade, here termed Parksosauridae, though it now also contains Asian and South American taxa that were not included as terminal taxa by *Boyd et al. (2009)*. The recovery of a monophyletic Parksosauridae significantly reduces the length of the inferred ghost lineages of many of its constituent members. For example, *Thescelosaurus neglectus* was once inferred to possess one of the longest ghost lineages in all of Dinosauria (~105 myr: *Weishampel & Heinrich, 1992*). Based on its position in the strict consensus tree, the inferred ghost lineage for this taxon is reduced by more than two-thirds (Fig. 5). Thus, not only is Parksosauridae well-supported by the character evidence, it also greatly improves the stratigraphic congruence of that subsection of the tree topology. However, a sizeable ghost lineage still exists at the base of Parksosauridae, extending ~40 myr from the Early Cretaceous back into the Middle Jurassic (Fig. 5). This implies that there is still much to learn regarding the early evolution and diversification of parksosaurids.

Recent analyses of the relationships of basal neornithischian taxa from Asia (e.g., *Changchunsaurus parvus, Haya griva*) have shown some support for a clade of basal neornithischian taxa endemic to Asia (*Butler et al., 2011; Makovicky et al., 2011; Han et al., 2012*). In its most inclusive form (*Makovicky et al., 2011; Han et al., 2012*) this clade consists of *Haya griva* as the sister taxon to a subclade composed of *Jeholosaurus shangyuanensis* + *Changchunsaurus parvus*. The exact position of this clade within Ornithischia is unresolved in the strict consensus trees of both *Butler et al. (2011)* and *Makovicky et al. (2011)*, though the maximum agreement subtrees presented by both authors place this clade near the base of Ornithopoda and the strict reduced consensus tree of *Han et al. (2012)* produces a similar result. However, *Makovicky et al. (2011)* cautioned that character support for these relationships was weak and that the large number of homoplastic characters displayed by these three taxa hinted at their possibly paraphyly. A different set of relationships is recovered for these taxa in this analysis. A clade consisting of *C. parvus* and *H. griva* is situated at the base of Thescelosaurinae within Parksosauridae (Fig. 2), while *J. shangyuanensis* is positioned outside of Parksosauridae near the base of Neornithischia. Given the incongruence between the results presented here and those of prior studies (e.g., *Butler et al., 2011; Makovicky et al., 2011; Han et al., 2012*), a brief discussion of the characters supporting the placement of these taxa in the present analysis is warranted.

This study incorporates new character data for *J. shangyuanensis* based on personal examination of multiple articulated and nearly complete specimens in the collections at Peking University, allowing much of the postcranial skeleton to be analyzed for the first time and for a clearer understanding of the cranial anatomy to be achieved (see Table 2 for a list of specimens examined). As a result, a set of key differences between *J. shangyuanensis* and the Asian taxa *C. parvus* and *H. griva* were noted that are crucial to determining the position of these taxa within Neornithischia. The ventral process of the predentary of *J. shangyuanensis* is unilobate (72:0), while in *C. parvus* and *H. griva* it is bifurcate (72:1). Six premaxillary teeth are present in *J. shangyuanensis* (112:0) in contrast to *C. parvus* and *H. griva* that display five premaxillary teeth (112:1). The morphology of the dentary of *J. shangyuanensis* is distinctly different than that of *C. parvus* and *H. griva*. In *J. shangyuanensis*, the anterior tip of the dentary is positioned close to the ventral margin (74:2), the ventral and dorsal margins of the dentary converge anteriorly (75:0), and the
dorsoventral height of the dentary just anterior to the coronoid process is less than 20% of the total length of the dentary (77:0). Alternatively, in *C. parvus* and *H. griva* the anterior tip of the dentary is positioned at midheight (74:1), the ventral and dorsal margins of the dentary are subparallel (75:1), and the dorsoventral height of the dentary just anterior to the rising coronoid process is greater than 20% of the total length of the dentary (77:1). Additionally, the crowns of the dentary teeth in *J. shangyuanensis* lack a prominent primary ridge (139:0), while a primary ridge is present on the dentary crowns of both *C. parvus* and *H. griva* (139:1). In the postcranial skeleton, the lateral surface of the greater trochanter of the femur is convex in *J. shangyuanensis* (213:0), while the lateral surface of the greater trochanter of the femur is flattened in *C. parvus* and *H. griva* (213:1). That character is an unambiguous synapomorphy of Parksosauridae, clearly indicating *C. parvus* and *H. griva* are parksosaurids, while *J. shangyuanensis* is positioned outside of this clade. Overall, the character evidence outlined above strongly argues against a close relationship between *J. shangyuanensis* and either *C. parvus* or *H. griva*.

Prior investigations into the systematic relationships of South American taxa previously referred to either Hypsilophodontidae (e.g., *Notohypsilophodon comodorensis*) or Iguanodontia (e.g., *Anabisetia saldiviai*) tended to be relatively restricted in scope, focusing largely on South American taxa (e.g., *Coria, 1999*; *Novas, Cambiaso & Ambrosio, 2004*; *Calvo, Porfiri & Novas, 2007*; *Rozadilla et al., 2016*). Those investigations often recovered South American taxa in an endemic clade (*Coria, 1999*; *Calvo, Porfiri & Novas, 2007*; *Rozadilla et al., 2016*), or closely situated to one another as part of a South American 'grade' of taxa (*Novas, Cambiaso & Ambrosio, 2004*). Several studies discussed tentative character support for some or all of these taxa forming a clade of strictly South American or Gondwanan taxa (*Coria & Calvo, 2002*; *Novas, Cambiaso & Ambrosio, 2004*; *Calvo, Porfiri & Novas, 2007*; *Ibiricu et al., 2010*). Thus, the recovery in this study of an iguanodontian clade comprised entirely of Gondwanan taxa is not unexpected. In fact, *Coria (1999)* previously recovered a clade composed of *Gasparinisaura cincosaltensis* + *Anabisetia saldiviai*, the same two South American taxa recovered as a part of this Gondwanan clade. *Coria (1999)* also suggested that *G. cincosaltensis* and *A. saldiviai* had evolved from other Gondwanan taxa and likely dispersed into South America via Antarctica, possibly from Australia, prior to the Cretaceous (*Coria, 1999*:57). The structure of the strict consensus tree obtained by this study (Fig. 2), the results of the biogeographic reconstructions (Figs. 3 and 4), the inferred distribution of ghost lineages for the taxa recovered within the Gondwanan clade (Fig. 5), and the recent discovery of iguanodontian taxa from Antarctica (*Morrosaurus antarcticus* and *Trinisaura santamartaensis*: *Coria et al., 2013*; *Rozadilla et al., 2016*) that appear to share a close relationship with several South American iguanodontian taxa (e.g., *Rozadilla et al., 2016*) all support this interpretation.

No prior analysis recovered a close relationship between any South American and Laurasian taxa, though some authors have suggested certain South American taxa more closely resembled Laurasian taxa than other Gondwanan taxa (e.g., *Talenkauen santacrucensis*; *Novas, Cambiaso & Ambrosio, 2004*). The placement of the South American taxa *Macrogryphosaurus gondwanicus*, *Notohypsilophodon comodorensis*, and *Talenkauen santacrucensis* within Thescelosaurinae amongst the North American taxa *Thescelosaurus*

and *Parksosaurus warreni* provide insight into the evolution of the ornithischian fauna of South America. The South American ornithischian taxa treated in this study are supported as parts of two distinct radiations that dispersed into South America at different times via separate geographic paths. Based on the results presented in this study, basal iguanodontian taxa dispersed into South America from Australia (possibly via Antarctica) during the Late Jurassic or the beginning of the Early Cretaceous (Figs. 4 and 5). Alternatively, thescelosaurine taxa most likely dispersed into South America from Asia (via North America) sometime during the latter portion of the Early Cretaceous, and then diversified, giving rise to Elasmaria (Figs. 4 and 5).

The close relationship between the South American members of Elasmaria and the North American taxa *Thescelosaurus* and *Parksosaurus warreni* may also answer some questions regarding the known stratigraphic distribution of parksosaurid taxa in North America during the Cretaceous. During most of the Cretaceous, orodromine taxa were the dominant basal neornithischian taxa present in North American faunas (*Sues, 1980*; *Scheetz, 1999*; *Weishampel et al., 2004*; *Varricchio, Martin & Katsura, 2007*; *Krumenacker, 2010*; *Brown et al., 2013*; *Gates et al., 2013*). At the end of the Campanian, all orodromine taxa disappear from the North American fossil record. In the Maastrichtian the thescelosaurine taxa *Thescelosaurus* and *Parksosaurus warreni* appear in the North American fossil record, which may be an example of faunal replacement (*Boyd et al., 2009*; *Brown, Boyd & Russell, 2011*). The results of the LFR biogeographic analysis suggest that the lineages leading to these latter two taxa may have originated in South America, and then dispersed into North America during the Maastrichtian. This observation strengthens the paleontological support for the presence of a land bridge and associated faunal interchange between North and South America during the latest Cretaceous (*Brett-Surman & Paul, 1985*; *Rage, 1986*; *Hutchinson & Chiappe, 1998*; *Ezcurra & Agnolin, 2011*).

## CONCLUSION

Our understanding of the basal relationships within Ornithischia and amongst the major ornithischian sub-clades is improving, aided by the discovery and description of new taxa, redescripion and re-evaluation of previously named taxa, and the implementation of improved phylogenetic methods and practices. While the phylogenetic hypothesis presented in Fig. 2 is among the best resolved, most stratigraphically consistent hypotheses yet proposed for this clade, additional improvements are needed. Clarification of the possible synonymization of *Lesothosaurus diagnosticus* and *Stormbergia dangershoeki* will help resolve questions regarding the split between and early evolution within the clades Thyreophora and Neornithischia. While support is building for the recognition of a clade of 'hypsilophodontids' that does not include *Hypsilophodon* (=Parksosauridae), the exact position of that clade relative to Marginocephalia and its taxonomic contents still fluctuates between analyses. While repositioning Parksosauridae within Ornithopoda substantially increases the length of the strict consensus tree obtained in this study, the future inclusion of additional character information, in terms of new characters, character state observations, and/or terminal taxa, will more robustly test this placement.

Additionally, while the current placement of Parksosauridae does decrease the ghost lineages for several parksosaurid taxa (e.g., *Thescelosaurus*), the ghost lineage for the base of Parksosauridae is still the longest within Ornithischia, indicating there is much regarding the early evolution within Neornithischia yet to learn. Finally, the complete absence of thescelosaurines and the dominance of orodromines in North America during most of the Cretaceous, followed by the abrupt disappearance of orodromines and wide geographic distribution of thescelosaurines in North America during the Maastrichtian remains an interesting puzzle. Increased North American sampling will help to further clarify if this temporal segregation is a real pattern or an artifact of sampling/preservation and additional work on non-cerapodan neornithischians from South America will provide additional information regarding biogeographic dispersals of thescelosaurines during the Cretaceous.

Given the differences between the phylogenetic hypothesis presented in this study and those presented by *Butler, Upchurch & Norman (2008)*, the natural next step is to work towards combining the character state observations presented in those datasets together (i.e., combining congruent characters, reaching a consensus on the description and scoring of character states, and removing or modifying characters and character states that are demonstrated to be influenced by tokogenetic and ontogenetic processes). The resulting dataset, possibly supplemented with additional new characters, would allow for these alternate hypotheses to be compared against a new phylogenetic hypothesis that considers all of the evidence put forth by those two studies. Such a dataset would not only be the most comprehensive test yet of basal ornithischian relationships, but would also form an excellent starting point for constructing a single dataset aimed at testing the evolutionary relationship of all ornithischian dinosaurs at the species level. Achieving such a goal will require the cooperation of numerous researchers, but would allow ornithischian relationships to be tested in such a way that all available character evidence is brought to bear on every evaluation of systematic relationships and ensure that homology statements within Ornithischia remain consistent across analyses. It would also guarantee that analyses of taxa positioned higher up the ornithischian tree will be properly rooted and character states correctly polarized. In the current technological age where data sharing and collaboration are easier than ever and analysis of large datasets is easily accommodated by a variety of search methods, the development of separate, competing phylogenetic datasets for the same taxonomic group should be abandoned and the construction of large scale datasets that include all available character observations is a goal that should be embraced by the entire research community.

## Institutional Abbreviations

| | |
|---|---|
| **BYU** | Brigham Young University, Provo, Utah, USA |
| **IVPP** | Institute of Vertebrate Paleontology and Paleoanthropology, Beijing, People's Republic of China |
| **LACM** | Natural History Museum of Los Angeles County, Los Angeles, California, USA |
| **MOR** | Museum of the Rockies, Bozeman, Montana, USA |

| | |
|---|---|
| **NCSM** | North Carolina Museum of Natural Sciences, Raleigh, North Carolina, USA |
| **NMV** | National Museum of Victoria, Melbourne, Australia |
| **PU** | Peking University, Beijing, People's Republic of China |
| **PKUP** | Peking University, Beijing, People's Republic of China |
| **ROM** | Royal Ontario Museum, Toronto, Ontario, Canada |
| **RSM** | Royal Saskatchewan Museum, Regina, Saskatchewan, Canada |
| **SAM** | Iziko South African Museum, Cape Town, South Africa |
| **SDSM** | South Dakota School of Mines and Technology, Rapid City, South Dakota, USA |
| **TMM** | Texas Memorial Museum, Austin, Texas, USA |
| **UMNH VP** | Utah Museum of Natural History, Salt Lake City, Utah, USA |
| **USNM** | Smithsonian Institution, National Museum of Natural History, Washington, D.C., USA |
| **UW** | University of Wyoming, Geological Museum, Laramie, Wyoming, USA |
| **YPM** | Yale Peabody Museum, New Haven, Connecticut, USA |

## ACKNOWLEDGEMENTS

Thank you to the members of my dissertation committee, J Clarke, C Bell, D Cannatella, P Makovicky, and T Rowe, for provided me with valuable guidance during the course of my graduate studies that greatly improved the quality of my research. I would also like to thank the former members of my committee at North Carolina State University (J Hibbard, M Schweitzer, and B Wiegmann) for their guidance and efforts on my behalf during my time at NCSU, which was very much appreciated.

Access to collections and specimens was generously provided by C Mehling and M Norell (AMNH), X Xing (IVPP), J Horner and B Baziak (MOR), V Schneider and D Russell (NCSM), KQ Gao (Peking University), S Shelton and M Greenwald (SDSM), J Nelson (TLAM), M Carrano and M Brett-Surman (USNM), S Sampson, M Getty, and R Irmis (UMNH), and D Brinkman and C Norris (YPM). P Brinkman (NCSM) provided guidance and collaborated in the preparation of NCSM 15728 and M Brown (VPL) provided instruction in the preparation of several new specimens of *Scutellosaurus*. D Evans (ROM) provided information on the anatomy of ROM 804. DA Winkler provided anatomical information about the 'Proctor Lake ornithopod.' D Varricchio provided additional information on the anatomy of MOR 1636a. CM Brown provided photographs of several specimens referred to *Thescelosaurus*, information regarding orodromine material from the Dinosaur Park Formation, and details regarding the anatomy of *Parksosaurus*. PM Galton provided encouragement and enlightening conversations over the years regarding basal ornithischian taxa. TP Cleland provided select photographs of NCSM 15728 and USNM 7758. S Nesbitt provided access to photographs of several basal ornithischian taxa from South Africa. I am indebted to S Masters for bringing the basal neornithischian material from the Kaiparowits Formation of Utah to my attention and A Titus of the Utah Bureau of Land Management for assistance in locating additional specimens referable to that new taxon. The Willi Hennig Society provided free access to the phylogenetic program TNT. The Polyglot Paleontologist website provided access to multiple translations of important

scientific research papers on ornithischian dinosaurs. ADB Behlke, TP Cleland, A DeBee, DR Eddy, T Gates, JL Green, ML Householder, J Hutchinson, EA Johnson, D Ksepka, K Lamm, S Nesbitt, K Pickett, D Pol, R Scheetz, NA Smith, HD Sues, editor J Anquetin, and reviewers P Makovicky and A McDonald provided thoughtful comments on previous drafts of my dissertation and this manuscript that greatly improved the final product.

### Funding

This research was funded via the American Museum of Natural History Collections Study Grant, an Ernest L. and Judith W. Lundelius Scholarship in Vertebrate Paleontology, a Francis L. Whitney Endowed Presidential Scholarship from the University of Texas, a Geological Society of America Graduate Student Research Grant, the National Science Foundation's East Asia and Pacific Summer Institutes for US Graduates Students program, the North Carolina Fossil Club. Additional financial support was received from the Jackson School of Geosciences at the University of Texas at Austin, the Department of Marine, Earth, and Atmospheric Sciences at North Carolina State University, and the Haslem Postdoctoral Fellowship at the South Dakota School of Mines & Technology. The funders had no role in study design, data collection and analysis, decision to publish, or preparation of the manuscript.

### Grant Disclosures

The following grant information was disclosed by the author:
American Museum of Natural History Collections Study Grant.
Ernest L. and Judith W. Lundelius Scholarship in Vertebrate Paleontology.
Francis L. Whitney Endowed Presidential Scholarship.
Geological Society of America Graduate Student Research Grant.
National Science Foundation's East Asia and Pacific Summer Institutes for US Graduates Students program.
North Carolina Fossil Club.
Jackson School of Geosciences at the University of Texas at Austin.
Department of Marine, Earth, and Atmospheric Sciences at North Carolina State University.
Haslem Postdoctoral Fellowship at the South Dakota School of Mines & Technology.

### Competing Interests

The author declares there is no competing interests.

### Author Contributions

- Clint A. Boyd conceived and designed the experiments, performed the experiments, analyzed the data, contributed reagents/materials/analysis tools, wrote the paper, prepared figures and/or tables, reviewed drafts of the paper.

### Data Availability

    The data set is provided as one of the supplementary tables (Table S3).
## Supplemental Information

Supplemental information for this article can be found online at http://dx.doi.org/10.7717/peerj.1523#supplemental-information.

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
