# Peer review of "The systematic relationships and biogeographic history of ornithischian dinosaurs"

_PeerJ, doi:10.7717/peerj.1523_

## Round 0.1 · original submission · Minor Revisions

· Academic Editor

Minor Revisions

The two reviewers agree on the quality of your submission and have only suggested minor revisions. Congratulations on maintaining such a high quality throughout this long manuscript. Please consider the reviewers’ comments in the revised version of your paper.

In addition to the suggestions from the reviewers, I have noted some minor issues that I list below:

- l. 53: you probably mean ‘Zheng et al., 2012’ instead of ‘Zhang et al., 2012’
- Institutional Abbreviations: If I am correct, several abbreviations (AMNH, CMN, KDRC, LACM, SDSM, YPM) are not cited in the text and tables, or only just in the Aknowledgements. In contrast, several abbreviations are cited, notably in Table 2, and missing from the list (BYU, SAM, PU, UW, ROM, TMM).
- l. 223: Boyd, 2012 is not referenced. Is it Boyd, 2014?
- l. 228 (and several other places): Huh et al. is dated from 2011 in the text, but 2010 in the reference list. Which one is correct?
- l. 323: ‘Parks’ instead of ‘Park’
- l. 411: Butler et al., 2008a or 2008b?
- l. 510: Zhao et al., 1999 not in the reference list
- l. 564: ‘Irmis et al., 2007’ instead of ‘Irmis et al., 2007b’
- l. 569: Coria et al., 2011 not in the reference list
- l. 660: Benton, 1994 not in the reference list
- l. 991: Sterling et al, 2010 not in the reference list
- l. 1166: Butler et al., 2008a or 2008b?
- Reference list: if I am correct, several references are not cited in the text or tables: Butler et al., 2012; Godefroit et al., 2009; Gradstein et al., 2005; Martin, 2009; Rauhut, 2003; Owen, 1942)
- Supplementary Table 2: Please include abbreviations in the Word document (easier for the reader).

·

Basic reporting

No Comments

Experimental design

No Comments

Validity of the findings

Results of Phylogenetic Analysis

Much of the focus here is, quite rightly, on the novel results regarding Parksosauridae and the “Gondwanan clade” of basal iguanodontians. However, the recovered topology among the representative members of Marginocephalia is highly unusual, with Archaeoceratops as sister taxon to the pachycephalosaur Wannanosaurus, and Yinlong and Liaoceratops as sister taxa.

These relationships are almost certainly due to undersampling. Even excluding the novel “Gondwanan clade”, there are 14 iguanodontians in the analysis, and only six marginocephalians.

I understand that this analysis has been in development for some time and that adding numerous new taxa at this stage is not wholly practical. However, I think the author should acknowledge the unusual results in Marginocephalia, and add more marginocephalians to future iterations. I recommend including a more complete pachycephalosaur, such as Stegoceras, as well as a broader sample of ceratopsians, such as Psittacosaurus, Mosaiceratops, and well known representatives of the three major Late Cretaceous clades Leptoceratopsidae, Protoceratopsidae, and Ceratopsoidea (see Farke et al. 2014 and Zheng et al. 2015 for recent analyses). This would mirror the author’s use of a range of iguanodontians, including rhabdodontids, dryosaurids, a basal ankylopollexian, and two basal hadrosauriforms.

Additional comments

This is an excellent paper that sets a very high standard for future work on ornithischian phylogeny and biogeography. It is commendable that the author made such an effort to include a variety of taxa that have been excluded from such analyses in the past. This will be a paper that other ornithischian scholars, such as myself, will cite heavily and find very useful.

There are some minor issues that need to be addressed.

Taxa of Interest

In the section on Agilisaurus, Yandusaurus hongheensis is said to be from the Middle Jurassic lower part of the Shaximiao Formation. In the section on Y. hongheensis itself, it is correctly said to be from the Upper Jurassic upper part of the formation (see Barrett et al. 2005).

Atlascopcosaurus was considered a nomen dubium by Agnolin et al. (2010). This paper also presented new data on Qantassaurus, Leaellynasaura, and Fulgurotherium.

It should be mentioned that Notohypsilophodon was recently redescribed by Ibiricu et al. (2014).

Parksosaurus is from the Tolman Member of the Horseshoe Canyon Formation in the Edmonton Group (Larson et al. 2010; Eberth and Braman 2012); the term “Edmonton Formation” is no longer used.

Yueosaurus was included in the analysis of Han et al. (2012).

Eousdryosaurus (Escaso et al. 2014) might be another useful taxon to include in a later version of this analysis, as it is the only Late Jurassic European dryosaurid.

The species name of Fulgurotherium australe is misspelled.

Nanosaurus agilis was considered diagnostic by Galton (2007).

The recently-named Morrosaurus antarcticus (Rozadilla et al. 2016) needs to be discussed, either as a taxon that is too fragmentary to be included or as an addition for a later version.

Figure 5

Several age ranges need to be adjusted in this figure:

The age of the Daohugou Beds in which Tianyulong was found is now thought to be Middle Jurassic (~164 Ma) (Gao and Shubin 2012, PNAS).

Yandusaurus is shown as being the same age as Agilisaurus and Hexinlusaurus, but it is actually younger (Barrett et al. 2005).

The age ranges of Iguanodon and Valdosaurus are too long. Iguanodon bernissartensis is known from a narrow interval in the Barremian-early Aptian (Norman 2014), and Valdosaurus canaliculatus is known only from the Barremian Wessex Formation (Barrett et al. 2011).

·

Basic reporting

This manuscript fulfills all criteria for publication set forth by PeerJ

Experimental design

The manuscript addresses a number of major questions in dinosaur evolutionary history with an unprecedented amount of data. All analyses are appropriate, properly described, and rigorous, and the results represent a significant advancement of knowledge.

Validity of the findings

As stated, this manuscript has amassed more data than other previous attempts to answer the same set of questions. The data is available in standard tables/ matrices. The discussion and conclusions are supported by the results, and tempered to identify areas of future improvement.

Additional comments

This manuscript represents a very useful contribution to our knowledge of dinosaur evolution, and will be widely cited (including by me), so I look forward to its publication. I find no major substantive concerns with the MS, and have only minor comments that I list by line number below. This manuscript does not require another round of review.

Line 53: Cite Makovicky et al., 2011 for new taxa from Asia

Line 76: the official title for LACM is Natural History Museum of Los Angeles County

Line 221: correct citation year for Gates et al. (2013)

Line 225: change "pes" to plural "pedes" or "feet"

Line 288: place a comma between "Late Cretaceous, and "upper"

Line 349: change "Upper" to lower case

Line 438: In Butler et al. (2008) Yandusaurus falls within a big polytomy that includes Marginocephalia in the strict consensus tree, and is only within Ornithopoda in the 50% Majority Rule tree. May be more accurate to state it was recovered as a neornithischian.

Lines 661&662: Why is GER* not considered? Its exclusion, whether for substantive or technical reasons, merits a comment.

Line 694: It would be more accurate to say this study "provides a comprehensive framework" as the adjective "robust" could be misinterpreted as qualifying all recovered relationships as well supported. Some are, but many aren't.

Line 817: Replace "Alternatively" with "Complementarily"

Line 868: I guess you mean "pubis" instead of "ischium"?

Line 964: by "inconspicuous" do you mean absent or highly reduced?

Line 977:"and" should not be italicized

Lines 1300-1301: Your study included an evaluation of the Butler et al. (2008) character set, so what specifically meant by combining the data? Consensus on character state definitions and how to score them are a big source of disagreement between phylogenies, so I am assuming this is a call to find common ground on this point?

Figs 3 & 4: Adding clade names will make it easier to keep track of the biogeographic results and discussion.

Table 2: Change "Kaiparowits hypsilophodontid" to "Kaiparowits orodromine"

Table 2: Zephyrosaurus appears to be missing from the table

---

## Round 0.2 · accepted · Accept

· Academic Editor

Accept

Thank you for the prompt reply. You have conscientiously followed the suggestions made by the reviewers and myself. I am therefore happy to inform you that your paper is now accepted for publication in PeerJ. Thank you for choosing us.

I have just noted very minor typos in two places in this revised version: a few spaces are missing in the first part of the abstract and ‘gaurenee’ should be changed to ‘guarantee’ in line 1383. Please, remember to include these changes at proof stage.